# MFN: Metadata-Free Real-World Noisy Image Generation

## Abstract

Real-world noise poses a significant challenge in signal processing, especially for denoising tasks. Although end-to-end denoising approaches have achieved exceptional performance, they are constrained to scenarios with abundant noisy-clean image pairs, which can be technically challenging and resource-intensive to collect. To address this issue, several generative methods have been developed to synthesize realistic noisy images from limited real-world datasets. While prior studies require camera metadata during training or testing to handle various real-world noise, the absence of metadata or variations in the information across different capturing devices is common in real-world scenarios, such as medical or microscope imaging, which limits their applicability. Thus, we aim to eliminate the need for explicit camera-related labels in both stages, enhancing applicability in real-world scenarios. To achieve this, we propose a novel framework called the Metadata-Free Noise Model (MFN), which extracts prompt features that encode input noise characteristics and generates diverse noisy images that adhere to the distribution of the input noise. Extensive experimental results demonstrate the superior performance of our model in real-world noise generation and denoising across various benchmark datasets.

## 1 Introduction

Real-world denoising is particularly challenging in low-level vision tasks due to the inherent complexity and diversity of noise. Unlike additive white Gaussian noise (AWGN), which is introduced in controlled environments, real-world noise originates from various sources, including camera sensor limitations, environmental conditions, and camera settings. Thus, denoising in real-world scenarios is more complex than in AWGN, where noise characteristics can be statistically modeled.

One clear method to address real-world denoising is to acquire a large dataset of noisy-clean image pairs and train denoising networks in a supervised manner (Zamir et al., 2020b; Chen et al., 2021; Zamir et al., 2022; Chen et al., 2022). However, this method presents a significant challenge: collecting a large-scale dataset is resource-intensive and technically complex (Abdelhamed et al., 2018; 2020; Xu et al., 2018; Nam et al., 2016; Plotz & Roth, 2017). To address this limitation, various methodologies for modeling real-world noise and generating datasets have been proposed (Abdelhamed et al., 2019; Kousha et al., 2022; Zamir et al., 2020a; Kim et al., 2024).

In this work, we aim to eliminate the reliance on metadata from noisy images when learning or generating them. Specifically, metadata, such as camera manufacturer and ISO settings, plays a crucial role as it offers a compact representation of how the ISP transforms the original RAW-RGB data, while distorting the corresponding noise. Several previous studies (Abdelhamed et al., 2019; Kousha et al., 2022; Fu et al., 2023) utilize metadata to guide the modeling of specific noise types; however, collecting or utilizing metadata in real-world scenarios can be challenging. In practical applications, such as medical imaging, the physical meaning of the information in the metadata is completely different or may even be unavailable. In these situations, conventional methods that depend on standardized metadata face significant limitations. To address this issue, we propose a novel framework, **M**etadata-**F**ree **N**oise model (**MFN**). This model extracts and exploits the underlying noise information in an unsupervised manner, without relying on metadata, while maintaining the quality of the generated samples.

Unlike the preceding approaches that rely on metadata during at least one phase of training or testing, illustrated as methods (a) (Abdelhamed et al., 2019; Kousha et al., 2022; Fu et al., 2023) and (b) (Kim et al., 2024) in Tab. 1, our method leverages prompt features that can replace the metadata to guide noise sampling. Specifically, we propose a **P**rompt **A**uto**e**ncoder (**PAE**) that encodes the noise of the input image and also produces input-specific prompt features including the characteristics of the input noise, such

Table 1: Difference between existing and proposed noise models.

| Methods | Metadata-Free? | |
|---|---|---|
| | Train | Test |
| (a) | ✗ | ✗ |
| (b) | ✗ | ✓ |
| Ours | ✓ | ✓ |

as ISO levels and noise correlation patterns. We then use a consistency model (CM) (Song et al., 2023; Luo et al., 2023), a type of generative model, to learn the latent space of the PAE and synthesize a new latent code conditioned on the produced prompt features which encapsulate information about the noise characteristics. Specifically, building on recent DiT (Peebles & Xie, 2023) architectures based on CM, we propose a **P**rompt **DiT** (**P-DiT**) that fully leverages the prompt features from the PAE during the generation process. Finally, we use the Decoder of our PAE to generate a noisy image by inputting the synthesized latent code from **P-DiT** along with the clean image. Through extensive experiments, we demonstrate that our MFN achieves outstanding real-world noise modeling quality without requiring any metadata during the training or testing phases. For downstream tasks, we trained a denoiser on the generated noisy dataset, achieving state-of-the-art (SOTA) denoising performance across various real-world benchmark datasets.

## 2 RELATED WORK

**sRGB Noise Generation.** The limited availability of real-world noise datasets results in overfitting issues for denoising networks, particularly in practical applications (Abdelhamed et al., 2019; Zamir et al., 2020a; Jang et al., 2021; Wu et al., 2023). To tackle this problem, various methods for generating sRGB noise have been proposed. PNGAN (Cai et al., 2021) treats each noisy pixel as a random variable, disentangling the noise generation problem into components in the image and noise domains. Flow-sRGB (Kousha et al., 2022) leverages normalizing flows to model noise distributions based on factors like smartphone type and gain settings. NeCA (Fu et al., 2023) introduces a neighboring correlation-aware noise model for synthesizing realistic noise, explicitly accounting for both signal dependency and neighboring noise correlations. However, these methods depend on explicit noisy image labels (*i.e.,* metadata) for noise generation, limiting their applicability in situations where such information is unavailable. To address this, NAFlow (Kim et al., 2024) introduces a noise-aware sampling algorithm to synthesize sRGB noise without requiring metadata during inference. However, it still requires metadata during the training phase. In contrast to these previous approaches, our approach offers a metadata-free noise generation method for both training and testing phases by leveraging prompts that capture input-noise-specific information.

**Prompt Learning.** Prompt-based methods are commonly employed in natural language processing (NLP) (Brown et al., 2020; Shin et al., 2020; Schick & Schütze, 2021) and vision tasks (Zhou et al., 2022a; Yao et al., 2024) to offer meaningful context for fine-tuning models on specific tasks. However, rather than relying on explicit manual guidance sets as prompts, recent methods (Zhou et al., 2022b; Potlapalli et al., 2023) have utilized learnable prompts to achieve more efficient parameter adaptation. While most prompting-based vision approaches focus on high-level tasks, recent low-level vision models (Li et al., 2023; Potlapalli et al., 2023; Wang et al., 2024) have been proposed to adaptively restore degraded images by leveraging their interactions with prompts. In particular, PromptIR (Potlapalli et al., 2023) and PromptRestorer (Wang et al., 2024) employ prompt learning to encode and implicitly classify degradation-specific information, enhancing the restoration model's performance. In this work, we develop a prompt learning method for realistic noisy image generation by introducing a Prompt Autoencoder (PAE) that extracts prompt features capturing the characteristics of the input noise.

## 3 PROPOSED METHOD

### 3.1 PRELIMINARIES

**Diffusion Models.** Diffusion models (Sohl-Dickstein et al., 2015; Song & Ermon, 2019; Ho et al., 2020) involve a forward process that corrupts data with independent and identically distributed Gaussian noise, followed by a learning phase to reverse this corruption. In the forward process, Gaussian

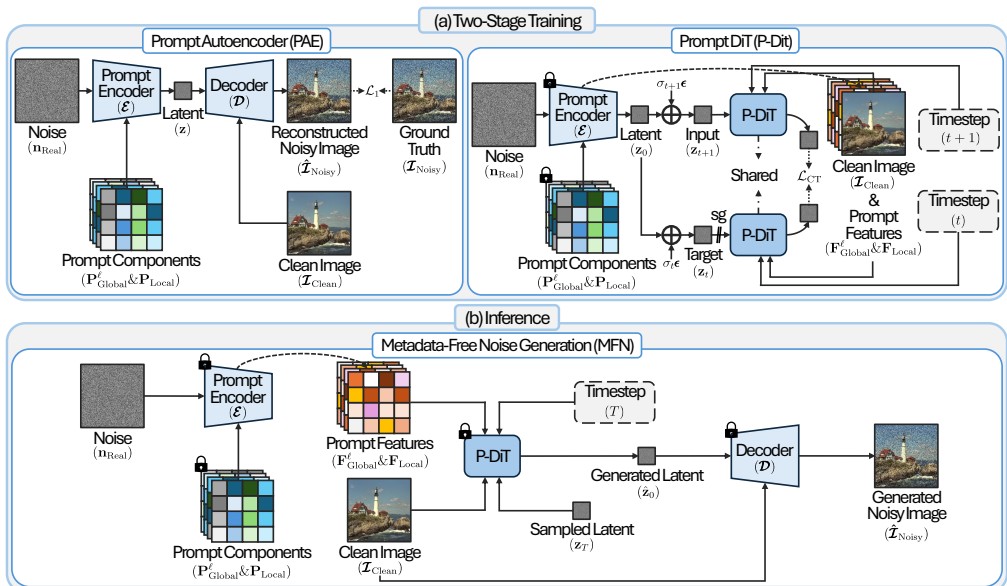

Figure 1: Overview of the proposed method. (a) Two-stage training phase. (b) Inference phase.

random noise is added to the given input $\mathbf{x}_0$. Using reparameterization, this process at timestep $t$ can be represented as:

$$\mathbf{x}_t = \alpha_t \mathbf{x}_0 + \sigma_t \boldsymbol{\epsilon}, \quad \boldsymbol{\epsilon} \sim \mathcal{N}(0, \mathbf{I}), \quad t \in \{0, 1, \dots, T\} \tag{1}$$

where $\mathbf{x}_t$ is the noise-added data sample at timestep $t$, $\mathbf{I}$ denotes the identity matrix, $\sigma_t$ is a noise schedule that controls the level of noise (*e.g.,* standard deviation of Gaussian noise), and $\alpha_t$ is a time-dependent function that define the specific type of diffusion process, such as the variance-preserving (Sohl-Dickstein et al., 2015; Ho et al., 2020) and variance-exploding (VE) (Song & Ermon, 2019; Song et al., 2021) diffusion processes (Kingma et al., 2021). In this work, we employ the VE diffusion process, where $\alpha_t$=1. The objective of diffusion models is to reverse the forward process by denoising the corrupted sample $\mathbf{x}_t$ back to the original data $\mathbf{x}_0$.

In the context of score-based generative models (Song & Ermon, 2019; 2020; Song et al., 2021), the score model $s_\theta(\mathbf{x}_t, \sigma_t)$ estimates the score function, a vector field that points toward regions of higher data density, by employing score matching (Hyvärinen & Dayan, 2005; Song et al., 2021). The reverse process is performed by iteratively applying the learned score function to gradually denoise the noisy input. This can be represented using probability flow ordinary differential equations (PF ODEs), as described in (Karras et al., 2022) as:

$$d\mathbf{x}_t = -\sigma_t \nabla_{\mathbf{x}_t} \log p(\mathbf{x}_t; \sigma_t) dt, \tag{2}$$

where the score function is $\nabla_{\mathbf{x}_t} \log p(\mathbf{x}_t; \sigma_t)$. In practice, solving this reverse-time ODE involves numerous update steps, which makes diffusion models computationally intensive.

**Consistency Models.** Unlike traditional diffusion models that generate data through multiple iterative steps, recent consistency models (CMs) streamline the process by achieving results in a single step (Song et al., 2023; Song & Dhariwal, 2024). Specifically, the CM learns a mapping function $f_\theta(\mathbf{x}_t, \sigma_t)$ such that, for any $\mathbf{x}_t$, the output remains *consistent* with the original data $\mathbf{x}_0$. This process can be approximated as follows:

$$f_\theta(\mathbf{x}_t, \sigma_t) = \mathbf{x}_t + \int_{\sigma_t}^{\sigma_0} \frac{d\mathbf{x}_u}{du} \, du \approx \mathbf{x}_0. \tag{3}$$

The consistency training (CT) loss ensures that the model's outputs remain stable across different noise levels, as follows:

$$\mathcal{L}_{\text{CT}} = \mathbb{E}\left[\lambda(\sigma_t) \, d\left(f_\theta(\mathbf{x}_{t+1}, \sigma_{t+1}) - \text{sg}(f_{\theta^-}(\mathbf{x}_t, \sigma_t))\right)\right], \tag{4}$$

where $\lambda(\cdot)$ is a weighting function, $d(\cdot)$ is a distance function like pseudo Huber loss (Song & Dhariwal, 2024), and sg$(\cdot)$ denotes the stop-gradient operator. Here, $f_\theta$ and $f_{\theta^-}$ refer to the student and teacher networks, respectively. The stop-gradient operator is used to keep the teacher network

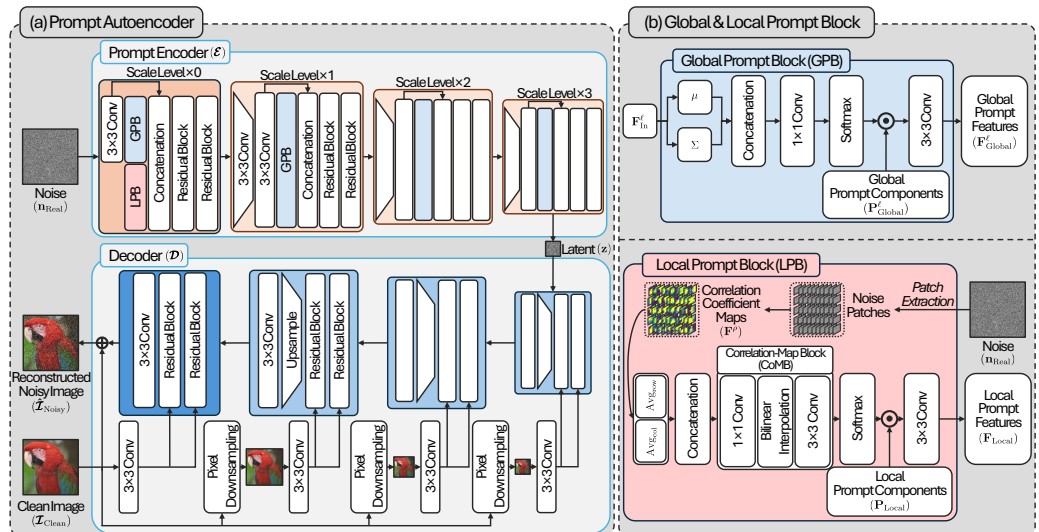

Figure 2: Sketch of the Prompt Autoencoder (PAE). (a) Architecture overview. (b) Details of Global and Local Prompt Block.

fixed during each optimization step of the student network. In our approach, since the pretrained teacher model is not available, we set $\theta^- = \theta$ (Song et al., 2023; Song & Dhariwal, 2024; Kong et al., 2024). We adopt most of the training hyperparameters for the CM from EDM (Karras et al., 2022) and iCT (Song & Dhariwal, 2024), and further details are in Sec. A.1.

## 3.2 OVERALL FLOW

We introduce a novel framework called Metadata-Free Noise (MFN), which can generate realistic noisy images without relying on explicit labels or metadata (*e.g.,* smartphone manufacturer, ISO setting). Our MFN is inspired by latent diffusion models (Rombach et al., 2022a; Luo et al., 2023), and its generation process operates within a compact latent space, further achieving speed improvements through CM. This approach significantly reduces computational and memory costs compared to methods that function in high-dimensional image space. We illustrate the overall flow of the MFN in Fig. 1, which comprises two main components: the Prompt Autoencoder (PAE) and the Prompt DiT (P-DiT), trained sequentially in a two-stage manner.

**Two-Stage Training Phase.** First, we train U-Net (Ronneberger et al., 2015) like PAE, which consists of Prompt Encoder $\mathcal{E}$ and Decoder $\mathcal{D}$, to learn a compact latent space. The Prompt Encoder $\mathcal{E}$ maps the input noise $\mathbf{n}_{\text{Real}}$, representing the residual between the paired noisy image $\mathcal{I}_{\text{Noisy}}$ and clean image $\mathcal{I}_{\text{Clean}}$, into a compact latent code $\mathbf{z}$, while also producing prompt features that capture the characteristics of the input noise. Given the clean image $\mathcal{I}_{\text{Clean}}$, the Decoder $\mathcal{D}$ reconstructs the noisy image from the encoded latent code.

Then, our P-DiT is trained to learn the distribution of the PAE's latent codes using a CM-based approach by optimizing $\mathcal{L}_{\text{CT}}$ in Eq. 4. Given the fully trained Prompt Encoder $\mathcal{E}$, our P-DiT learns to synthesize latent codes that embed input-specific noise characteristics by leveraging an encoded latent code $\mathbf{z}_0$ and prompt features from Prompt Encoder $\mathcal{E}$, along with the clean image.

**Inference Phase.** During the generation phase, the Prompt Encoder $\mathcal{E}$ produces prompt features that capture the noise characteristics given an input noise. Conditioning on these prompts and a clean image, the P-DiT generates a latent code $\hat{\mathbf{z}}_0$, which is then decoded by the Decoder $\mathcal{D}$ to produce the final noisy image $\hat{\mathcal{I}}_{\text{Noisy}}$ given the clean image $\mathcal{I}_{\text{Clean}}$.

## 3.3 PROMPT AUTOENCODER

### 3.3.1 PROMPT ENCODER

As depicted in Fig. 2, the Prompt Encoder $\mathcal{E}$ consists of multiple convolutional layers, such as residual blocks (He et al., 2016), incorporating two key components to extract attributes of the input noise: the Global Prompt Block (GPB) and the Local Prompt Block (LPB), both inspired by the

ISP pipeline. GPB and LPB generate prompt features that are concatenated with the input features to embed informative latent codes. Specifically, learnable prompt components in both blocks are trained to capture the dataset distribution through loss optimization. The resulting prompt features are dynamically adjusted based on real-world noise characteristics, such as ISO levels and noise correlation. Hence, these prompt features can act as implicit representations, capturing the input noise characteristics and thereby eliminating the reliance on explicit labels and metadata.

**Global Prompt Block.** A camera ISP can adjust brightness using ISO, simulating physical exposure. However, increasing the signal gain through high ISO settings also amplifies sensor noise. Thus, ISO is a crucial factor in modeling noise attributes. To capture global noise statistics, such as noise amplification in the prompt features, we propose the Global Prompt Block (GPB). This block is designed to analyze how different ISO levels impact noise characteristics, allowing our model to better represent the inherent noise patterns associated with varying ISO settings.

In the GPB, we first define a set of learnable global prompt components $\mathbf{P}_{\text{Global}}^{\ell} \in \mathbb{R}^{\frac{H}{2\ell} \times \frac{W}{2\ell} \times C_{\text{Global}}^{\ell}}$ to capture the global statistics at scale $\ell$. To generate the global prompt features $\mathbf{F}_{\text{Global}}^{\ell} \in \mathbb{R}^{\frac{H}{2\ell} \times \frac{W}{2\ell} \times C_{\text{Global}}^{\ell}}$, we compute input-specific coefficients $\mathbf{w}_{\text{Global}}^{\ell}$ from the input feature $\mathbf{F}_{\text{In}}^{\ell} \in \mathbb{R}^{\frac{H}{2\ell} \times \frac{W}{2\ell} \times C_{\text{In}}^{\ell}}$ by computing the channel-wise mean and standard deviation of the input feature to capture the global statistics of the input noise as:

$$\mathbf{w}_{\text{Global}}^{\ell} = \text{Softmax}\left( \text{Conv}_{1\times1}\llbracket \mu(\mathbf{F}_{\text{In}}^{\ell}), \Sigma(\mathbf{F}_{\text{In}}^{\ell}) \rrbracket \right), \tag{5}$$

where $\mu(\cdot)$ and $\Sigma(\cdot)$ represent functions that compute the channel-wise mean and standard deviation, respectively. These two moments are concatenated and passed through a $1 \times 1$ convolutional layer, followed by a softmax operation. Then, these coefficients are used to dynamically modify the global prompt components $\mathbf{P}_{\text{Global}}^{\ell}$, followed by a $3\times3$ convolutional layer for refinement, to yield the final global prompt features $\mathbf{F}_{\text{Global}}^{\ell}$ as:

$$\mathbf{F}_{\text{Global}}^{\ell} = \text{Conv}_{3\times3}\left( \mathbf{w}_{\text{Global}}^{\ell} \odot \mathbf{P}_{\text{Global}}^{\ell} \right), \tag{6}$$

where $\odot$ denotes the element-wise multiplication. This process ensures that the global prompt features reflect the global noise characteristics based on noise distribution statistics in the input image.

**Local Prompt Block.** In the GPB, we extract global information (*e.g.,* ISO); however, real-world sRGB noise cannot be fully characterized by global information alone. This issue arises from several transformations in the ISP pipeline, such as non-linear and locally different spatial operations, which cause patterned, non-IID, and signal-dependent noise during the RAW-to-sRGB conversion. Therefore, we additionally propose the Local Prompt Block (LPB) to capture capture camera model-specific and signal-dependent noise characteristics from the input noise.

In the LPB, we first extract $\rho\times\rho$ patch from the input noise $\mathbf{n}_{\text{Real}} \in \mathbb{R}^{H \times W \times 3}$ at every pixel location and calculate the local correlation map for each patch with respect to its center pixel. Specifically, following the approach in LGBPN (Wang et al., 2023), we compute the Pearson correlation coefficients of neighboring pixels relative to the center pixel for each patch. This operation is applied to all patches, resulting in correlation coefficient maps $\mathbf{F}^{\rho} \in \mathbb{R}^{H \times W \times \rho^2}$, where each pixel contains its local correlation information along the channel axis. We then separately compute row-wise and column-wise averages from $\mathbf{F}^{\rho}$ to capture local distortions introduced by the ISP pipeline within each patch. These two averages are then concatenated and upscaled using correlation map block (CoMB) which consists of lightweight operations, including a $1\times1$ convolutional layer, a bilinear upsampler, and a $3\times3$ convolutional layer, as illustrated in Fig. 2 (b). Similar to the GPB, we apply a softmax operation to the upscaled features to generate the local prompt coefficients $\mathbf{w}_{\text{Local}} \in \mathbb{R}^{H \times W \times C_{\text{Local}}}$ as:

$$\mathbf{w}_{\text{Local}} = \text{Softmax}\left( \text{CoMB}\left( \llbracket \text{Avg}_{\text{row}}(\mathbf{F}^{\rho}), \text{Avg}_{\text{col}}(\mathbf{F}^{\rho}) \rrbracket \right) \right), \tag{7}$$

where $\text{Avg}_{\text{row}}$ and $\text{Avg}_{\text{col}}$ indicate row-wise and column-wise averaging operations, respectively. Next, the local prompt coefficients are multiplied element-wise by the prompt components $\mathbf{P}_{\text{Local}} \in \mathbb{R}^{H \times W \times C_{\text{Local}}}$ for dynamic feature aggregation, yielding the local prompt features $\mathbf{F}_{\text{Local}} \in \mathbb{R}^{H \times W \times C_{\text{Local}}}$ as follows:

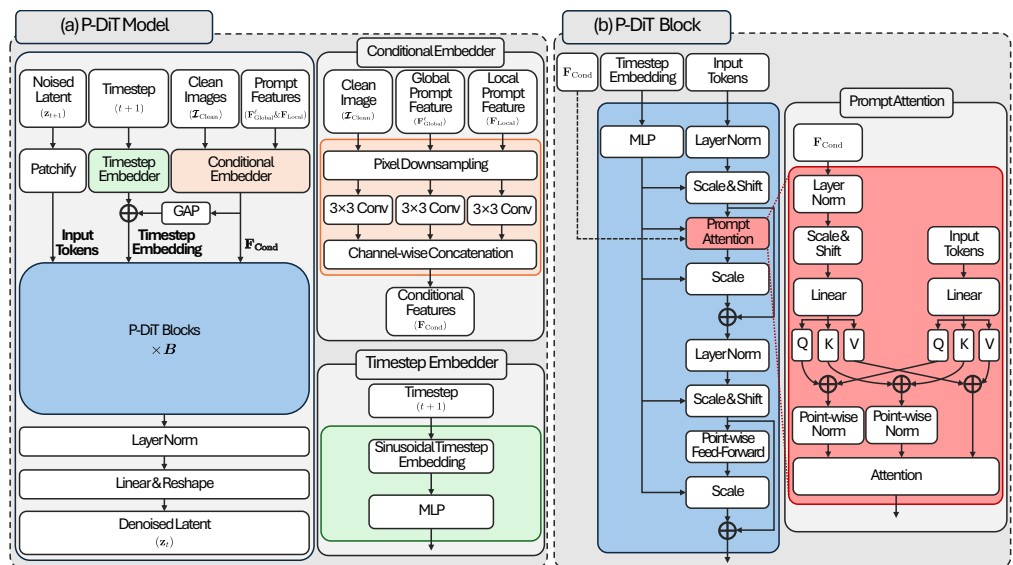

Figure 3: Overview of Prompt DiT (P-DiT). (a) Overall P-DiT structure. (b) P-DiT block.

$$\mathbf{F}_{\text{Local}} = \text{Conv}_{3\times3}\left(\mathbf{w}_{\text{Local}} \odot \mathbf{P}_{\text{Local}}\right). \tag{8}$$

We emphasize that by leveraging local noise characteristics (*e.g.,* noise local correlation), the prompt components learn to focus on unique camera-specific and signal-dependent features.

### 3.3.2 DECODER

As illustrated in the lower part of Fig. 2 (a), the Decoder $\mathcal{D}$ consists of numerous convolutional layers, integrating residual blocks and upsampling operations to transform latent codes back into noisy images. The upsampling operation increases the spatial resolution of input features at scale $\ell$ using nearest-neighbor upsampling, as followed by (Rombach et al., 2022b). To further capture the signal-dependent characteristics of real-world noise, clean images are downsampled using pixel downsampling (Shi et al., 2016), preserving fine-grained textures and conditioning at each scale level.

### 3.4 PROMPT DIT

In Fig. 3, we introduce P-DiT, which fully leverages the prompts extracted from the Prompt Encoder $\mathcal{E}$ to synthesize latent codes that align with the embedded information of the input noise characteristics. Our P-DiT is based on DiT (Peebles & Xie, 2023), a transformer-based CM architecture specifically designed for training diffusion models.

In Fig. 3 (a), the P-DiT consists of a series of $B$ P-DiT blocks through which the noise added input latent $\mathbf{z}_{t+1}$ is processed. Moreover, for conditioning features, we use the timestep $t+1$, clean images $\mathcal{I}_{\text{Clean}}$, and prompt features $\mathbf{F}_{\text{Local}}$ and $\mathbf{F}_{\text{Global}}^{\ell}$. The timestep enables the model to adjust its denoising predictions based on the noise levels of the diffusion process at each timestep. More precisely, the timestep is embedded by a timestep embedder, composed of sinusoidal embeddings and an MLP block (Vaswani, 2017). We also use clean images to help the model learn the signal-dependent properties of real-world noise and prompt features. This enables the generation of latent codes that capture input noise characteristics without relying on metadata, with these features embedded by a conditional embedder. Specifically, we first downsample these inputs to match the spatial size of the latent codes and then extract shallow features separately using $3 \times 3$ convolutional layers, which are concatenated along the channel axis as follows:

$$\mathbf{F}_{\text{Cond}} = \left[\!\left[\text{Conv}_{3\times3}\big(\text{PD}(\mathcal{I}_{\text{Clean}})\big), \text{Conv}_{3\times3}\big(\text{PD}(\mathbf{F}_{\text{Local}})\big), \text{Conv}_{3\times3}\big(\text{PD}(\mathbf{F}_{\text{Global}}^{\ell})\big)\right]\!\right], \tag{9}$$

where $\text{PD}(\cdot)$ indicates a pixel downsampling operator. Note that we use $\mathbf{F}_{\text{Global}}^{\ell}$ for all scale levels, and each feature is processed separately. The concatenated feature $\mathbf{F}_{\text{Cond}}$ is then processed with global average pooling and added to the time embeddings.

In Fig. 3 (b), we present the P-DiT block. We employ adaptive layer normalization (AdaLN) as the conditioning mechanism to modulate the statistics of input features. AdaLN consists of two components: layer normalization (Perez et al., 2018) and adaptive modulation. AdaLN first normalizes the input features to have a mean of zero and a standard deviation of one. Then, the normalized features are modulated using scale and shift parameters derived from the conditioning input.

Moreover, to fully leverage the available information from the conditions and capture locally varying spatial correlations and signal-dependent noise characteristics in the prompt features, we further enhance these features through a prompt attention mechanism, referred to as Prompt Attention.

**Prompt Attention.** As illustrated in Fig. 3 (b), we use the conditions to generate the key, query, and value within the attention layer, similar to MMDiT (Esser et al., 2024), enabling the model to effectively capture the spatial information of the prompt features. Specifically, we first modulate the combined conditional features $\mathbf{F}_{\text{Cond}}$ through AdaLN to extract time-dependent information. Then, we generate the key, query, and value features using a single linear layer and combine them with the corresponding features from the input tokens through element-wise addition. Using these conditioned features, we apply cosine attention (Luo et al., 2018; Karras et al., 2024) with point-wise normalization (Karras et al., 2018) to stabilize the training process (Karras et al., 2024).

## 4 EXPERIMENTS

Please refer to the appendix for detailed information and additional experimental results. The source code will also be made publicly available upon acceptance.

### 4.1 EXPERIMENTAL SETUP

**Implementation Details.** The PAE is trained with the Adam optimizer (Kingma & Ba, 2014) and minimize the $\mathcal{L}_1$ loss between the noisy image $\boldsymbol{\mathcal{I}}_{\text{Noisy}}$ and the reconstructed noisy image $\hat{\boldsymbol{\mathcal{I}}}_{\text{Noisy}}$, along with the $\mathcal{L}_2$ regularization applied to the latent code $\mathbf{z}$ for dense representation (Rombach et al., 2022b). We start with an initial learning rate 1e-4, which is then reduced to 1e-6 using a cosine annealing algorithm (Loshchilov & Hutter, 2017) over 400k iterations. We use randomly cropped patches of size $256{\times}256$ and a mini-batch size of 64 for training.

The P-DiT model is optimized with the RAdam optimizer (Liu et al., 2020) using a fixed learning rate of 2e-4 over 250k iterations, applying the pseudo-Huber loss for Eq. 4, similar to iCT (Song & Dhariwal, 2024). We also use randomly cropped patches of size $256{\times}256$ to embed latent codes of size $32{\times}32$, with a mini-batch size of 512 during training. For the ablations, the mini-batch size is reduced to 128 to save training time. More details are in Sec. A.1.

For denoising, we adopt the DnCNN architecture (Zhang et al., 2017)), optimized with the Adam optimizer (Kingma & Ba, 2014) over 100k iterations. The training follows the setup outlined in (Fu et al., 2023; Kim et al., 2024), with a learning rate of 1e-3 and a mini-batch size of 8.

**Dataset.** To train the PAE, P-DiT, and DnCNN, we employ the SIDD (Abdelhamed et al., 2018) dataset, which includes 34 different camera configurations. We adopt the SIDD Medium split, comprising 320 noisy-clean image pairs captured with five unique smartphone cameras: Google Pixel (GP), iPhone 7 (IP), Samsung Galaxy S6 Edge (S6), Motorola Nexus 6 (N6), and LG G4 (G4). To evaluate the generated noise quality, we use the SIDD validation set, SIDD+ (Abdelhamed et al., 2020), PolyU (Xu et al., 2018), and Nam (Nam et al., 2016) dataset, and the DND (Plotz & Roth, 2017) dataset is additionally employed to assess denoising performance. Note that the SIDD+, PolyU, Nam, and DND datasets were captured using different types of camera sensors.

**Metrics.** To examine the quality of the generated noise, we use two metrics: Kullback-Leibler Divergence (KLD) and Average Kullback-Leibler Divergence (AKLD) (Yue et al., 2020). Additionally, we employ PSNR and SSIM metrics to assess the performance of denoising.

### 4.2 QUANTITATIVE AND QUALITATIVE RESULTS ON NOISE GENERATION

**Device-Specific Noise Quality Assessment.** In Tab. 2, we evaluate the noise quality of each smartphone device type to assess device-specific noise generation performance in terms of KLD and AKLD. Our method is compared with four other models: C2N (Jang et al., 2021), Flow-sRGB (Kousha et al., 2022), NeCA-W (Fu et al., 2023), and NAFlow (Kim et al., 2024). To align

Table 2: Quantitative results for synthetic noise on a subset of the SIDD validation set, in which the ISO values overlap with the training set. The results are computed with KLD↓ and AKLD↓. The best and second-best results are shown in **bold** and underline.

| Camera | Metrics | C2N | Flow-sRGB | NeCA-W | NAFlow | **MFN** |
|--------|---------|------|-----------|--------|--------|---------|
| G4 | KLD | 0.1660 | 0.0507 | 0.0242 | 0.0254 | **0.0174** |
|    | AKLD | 0.2007 | 0.1504 | 0.1524 | 0.1367 | **0.1283** |
| GP | KLD | 0.1315 | 0.0781 | 0.0432 | 0.0352 | **0.0143** |
|    | AKLD | 0.1968 | 0.1797 | 0.1273 | 0.1180 | **0.1074** |
| IP | KLD | 0.0581 | 0.5128 | 0.0410 | 0.0339 | **0.0291** |
|    | AKLD | 0.2929 | 1.7490 | 0.1145 | 0.1522 | **0.1128** |
| N6 | KLD | 0.3524 | 0.2026 | 0.0206 | 0.0309 | **0.0167** |
|    | AKLD | 0.2919 | 0.2469 | 0.1304 | 0.1108 | **0.1106** |
| S6 | KLD | 0.4517 | 0.3735 | 0.0302 | 0.0272 | **0.0193** |
|    | AKLD | 0.4190 | 0.2641 | 0.1933 | 0.1355 | **0.1223** |
| Average | KLD | 0.2129 | 0.2435 | 0.0342 | 0.0305 | **0.0194** |
|         | AKLD | 0.2802 | 0.5180 | 0.1436 | 0.1306 | **0.1163** |

Table 3: Quantitative results of synthetic noise on the PolyU, Nam, SIDD validation set and SIDD+. All methods are trained with SIDD training set. The results are computed with KLD↓ and AKLD↓. The best results are shown in **bold**.

| Methods | PolyU | | Nam | | SIDD | | SIDD+ | | Average | |
|---------|-------|-------|-----|-------|------|-------|-------|-------|---------|-------|
|         | KLD↓ | AKLD↓ | KLD↓ | AKLD↓ | KLD↓ | AKLD↓ | KLD↓ | AKLD↓ | KLD↓ | AKLD↓ |
| NAFlow | 0.2304 | 1.7801 | 0.2055 | 1.9320 | 0.0291 | 0.1293 | 0.0494 | 0.2917 | 0.1286 | 1.0333 |
| Ours | **0.0822** | **0.5814** | **0.0966** | **0.4668** | **0.0189** | **0.1160** | **0.0283** | **0.1630** | **0.0565** | **0.3318** |

Figure 4: Visualization of synthetic noisy images on the SIDD validation set. From left to right: NeCA-W, NAFlow, Ours (MFN), and real noisy images.

with the experimental settings of Flow-sRGB, NeCA-W and NAFlow, we ensure that both the training and validation sets contain the same ISO levels.

Compared to other methods, our MFN achieves the best performance across all device types, demonstrating significant improvements over NAFlow (the state of the art at the time of submission) in both average KLD and AKLD scores. Unlike the other methods, our MFN does not require metadata during training or testing, highlighting its versatility in real-world scenarios. We also present visual comparisons of the noisy images generated by each method in Fig. 4. The MFN generates more natural and realistic noise that closely resembles real-world noise distributions in magnitude and correlation patterns, demonstrating its superior performance in generating realistic noise.

**Metadata-Free Noise Quality Assessment.** In Tab. 3, we validate the robustness of our method by evaluating the noise quality on external real-world datasets that were not used during the training phase. We compare our MFN with NAFlow, as both models generate input-specific noise without requiring explicit metadata at the inference phase. However, unlike our approach, NAFlow requires metadata for training. In addition to the entire SIDD validation set, we compare results on three external datasets: PolyU, NAM, and SIDD+. These datasets were captured using various device types, camera sensors, and ISPs (*e.g.,* smartphones and DSLRs).

The overall results demonstrate that the MFN consistently outperforms NAFlow across all four datasets, achieving substantial gains in average KLD and AKLD scores. This emphasizes the MFN's robustness across various camera settings and external datasets.

### 4.3 APPLICATION: REAL-WORLD DENOISING

**Denoising Results on SIDD.** To evaluate the efficacy of noise modeling methods, we train the DnCNN (Zhang et al., 2017) using synthetic noisy-clean paired datasets. In Tab. 4, we compare

Table 4: Quantitative results of denoising performance on SIDD-Benchmark in terms of PSNR↑ and SSIM↑. All methods are trained with synthetic noisy-clean pairs. Note that *Real* are trained with real noisy-clean pairs. The best and second-best results are highlighted in **bold** and underline.

| Metrics | C2N | NoiseFlow | Flow-sRGB | NeCA-W | NAFlow | MFN | *Real* |
|---|---|---|---|---|---|---|---|
| PSNR↑ | 33.76 | 33.81 | 34.74 | 36.82 | 37.22 | 37.55 | **37.63** |
| SSIM↑ | 0.901 | 0.894 | 0.912 | 0.932 | 0.935 | **0.937** | 0.936 |

Table 5: Quantitative results of denoising performance on PolyU, Nam, DND, and SIDD+ in terms of PSNR↑ and SSIM↑. All methods are trained with synthetic noisy-clean pairs of SIDD training set. The best and second-best results are highlighted in **bold** and underline.

| Methods | PolyU | | Nam | | DND | | SIDD+ | | Average | |
|---|---|---|---|---|---|---|---|---|---|---|
| | PSNR↑ | SSIM↑ | PSNR↑ | SSIM↑ | PSNR↑ | SSIM↑ | PSNR↑ | SSIM↑ | PSNR↑ | SSIM↑ |
| *Real* | 36.34 | 0.9204 | 35.35 | 0.8828 | **38.85** | 0.9434 | 35.68 | 0.8860 | 36.56 | 0.9082 |
| NAFlow | 36.85 | 0.9499 | 37.45 | 0.9524 | 30.95 | 0.8010 | **36.67** | **0.9249** | 35.48 | 0.9071 |
| Ours | **37.93** | **0.9609** | **38.08** | **0.9630** | 38.75 | **0.9468** | 36.23 | 0.9072 | **37.75** | **0.9445** |

Figure 5: Visual comparison on denoising results with PSNR↑ on SIDD validation set from DnCNN trained on each method. For more qualitative results, please refer to the appendix.

the denoising performance of our MFN with (Jang et al., 2021), Flow-sRGB (Kousha et al., 2022), NeCA-W (Fu et al., 2023), NAFlow (Kim et al., 2024), and *Real*. Notably, *Real* represents results from real (not synthetic) noisy-clean paired data, serving as an oracle for synthetic noise generation.

Our MFN outperforms other generative methods, surpassing NAFlow by over 0.33 dB in PSNR and 0.002 in SSIM. Moreover, the denoising results produced by MFN closely match those of Real, achieving a PSNR gap of only 0.08 dB and even exceeding SSIM by 0.001.

**Generalization on Various Real-World Datasets.** To further evaluate the generalization capabilities, we assess denoising performance on four additional benchmark datasets: PolyU, NAM, DND, and SIDD+, which feature a variety of noise patterns and image characteristics. The results in Tab. 5 summarize the comparison of our method with two approaches: NAFlow and *Real*, trained using synthetic and ground-truth noisy-clean pairs from the SIDD training set, respectively.

Our method achieved superior performance across most datasets, demonstrating the best averaged results with 37.75 dB in PSNR and 0.9445 in SSIM, outperforming both NAFlow and *Real*. While NAFlow shows the best denoising performance on the SIDD+ dataset, which was captured with smartphone cameras like SIDD (*i.e.,* training data), it has the lowest average score, indicating overfitting to the training data (SIDD) and poor generalization ability. These findings suggest that our method is highly effective for general denoising tasks, striking a better balance between noise removal and structural preservation across diverse real-world datasets.

## 4.4 METADATA CLASSIFICATION

We evaluate the extent to which the prompts extracted from the Prompt Encoder $\mathcal{E}$ capture the characteristics of the input noise.

First, we trained ResNet-based classifiers using either the input noise $\mathbf{n}_{\mathrm{Real}}$ or the prompt features as input, measuring their ability to accurately categorize camera sensor types (i.e., camera manufacturer). We selected five major camera sensors from the SIDD dataset, resulting in five distinct labels, denoted as *Camera Sensor* in the second column of Tab. 6. The classification results show that using features of GPB as input improves camera sen-

Table 6: Quantitative results of metadata classification on SIDD validation with different combinations of prompts in terms of accuracy↑. We test the validity of the prompts in two aspects: camera sensor type and ISO level. The best and second-best results are highlighted in **bold** and underline.

| Classifier | *Camera Sensor* (%) | *Camera Sensor + ISO Level* | |
|---|---|---|---|
| | | Top-1 (%) | Top-3 (%) |
| Baseline (No Prompt) | 75.80 | 66.27 | 93.91 |
| GPB | 82.37 | 68.35 | 94.71 |
| GPB + LPB | **94.47** | **75.48** | **98.64** |

Table 7: Effect of GPB and LPB on noise generation. The best results are shown in **bold**.

| Combination | | Generation | |
|---|---|---|---|
| GPB | LPB | KLD↓ | AKLD↓ |
| ✗ | ✗ | 0.6182 | 0.4387 |
| ✓ | ✗ | 0.0287 | 0.1112 |
| ✓ | ✓ | **0.0261** | **0.1108** |

Table 8: Effect of GPB and LPB on noisy image reconstruction. The best results are shown in **bold**.

| Combination | | Reconstruction | |
|---|---|---|---|
| GPB | LPB | PSNR↑ | SSIM↑ |
| ✗ | ✗ | 37.58 | 0.9800 |
| ✓ | ✗ | 46.30 | 0.9982 |
| ✓ | ✓ | **46.54** | **0.9983** |

Table 9: KLD score depending on different number of blocks $B$ in Fig. 3.

| # Blocks | KLD↓ |
|---|---|
| $B = 4$ | 0.0350 |
| $B = 6$ | 0.0345 |
| $B = 8$ | **0.0261** |

sor prediction accuracy compared to the baseline model, which uses real noise as input (without GPB and LPB). Furthermore, combining GPB with LPB further enhances classification accuracy. These results demonstrate that the prompts contain information about the camera sensor.

Next, we created a total of 16 labels by simultaneously considering these five sensors and additional ISO levels, and conducted classification experiments. We compared their classification performance on the SIDD validation set, measuring Top-1 and Top-3 accuracy (last two columns in Tab. 6). The results show that using both GPB and LPB achieves the best classification performance for identifying the camera sensor and ISO configuration, highlighting the effectiveness of the proposed prompt blocks in capturing input noise-dependent characteristics.

Further details regarding the classifier architecture and labels can be found in Sec. A.3.

## 4.5 Ablation Study

**Effect of GPB and LPB on Noise Generation.** Our PAE encodes input noise into a compact latent code and generates prompt features that capture input-specific noise characteristics. To evaluate the impact of the prompts extracted by the GPB and LPB, we conducted experiments, as shown in Tab. 7. The P-DiT model is differently trained with various PAE variants to investigate the effect of each prompt. Combining diverse information from the prompt blocks such as ISO settings and noise correlation enhances the quality of synthesized noise by providing vital input-specific distribution details.

**Effect of GPB and LPB on Noisy Image Reconstruction.** As described in Tab. 8, we also assessed the influence of both prompt blocks on noisy image reconstruction. Incorporating GPB and LPB enhances the fidelity of the reconstructed images, as the prompt features encompass meaningful information about noise characteristics at different scales (see Fig. 2). Therefore, it is essential to utilize both prompt features to enhance the quality of the reconstructed images. Notably, we pass the latent directly from the Prompt Encoder $\mathcal{E}$ to the Decoder $\mathcal{D}$, without using the P-DiT for this experiment.

**Effect of Number of P-DiT Blocks.** As shown in Tab. 9, we measured the KLD scores across different numbers of blocks $B$ in the P-DiT to evaluate their impact on noise generation, with $B$ representing the hierarchical levels used in our P-DiT. The results indicate that increasing the number of blocks improves the quality of noise generation. Even with half the number of blocks, P-DiT yet achieves a moderate KLD score, with only a slight difference from the final version ($B$=8), suggesting an opportunity to optimize the trade-off between model complexity and performance.

## 5 Conclusion

In this work, we propose a novel MFN framework to remove the reliance on metadata when generating real-world noise. Our MFN comprises two key components: Prompt Autoencoder (PAE) and Prompt DiT (P-DiT). The PAE encodes input noise, generating a compact latent code and extracting meaningful prompt features at various scales to replace input noise-specific metadata. The P-DiT then synthesizes latent codes using prompts and clean images as conditions. Finally, the generated latent codes are fed into the PAE Decoder alongside clean images to produce realistic noisy images. A key advantage of our method is that it does not require explicit metadata during either the training or testing phases unlike conventional methods, highlighting its strength in real-world applications. Moreover, experimental results on real-world benchmarks demonstrate that our framework excels in both real-world noise generation and downstream denoising tasks.

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

## A  APPENDIX

### A.1  TRAINING DETAILS OF P-DiT

#### A.1.1  CM PARAMETERIZATION

The CM-based model $f_\theta$ aims to approximate the consistency function $f(\cdot, \cdot)$, which satisfies $f(\mathbf{x}_t, \sigma_t) = \mathbf{x}_0$. Therefore, it must adhere to the boundary condition $f(\mathbf{x}_0, \sigma_0) = \mathbf{x}_0$. To ensure this, we follow the parameterization used in EDM (Karras et al., 2022) and CM (Song et al., 2023), defining the model as follows:

$$f_\theta(\mathbf{x}_t, \sigma_t) = c_{\text{skip}}(\sigma_t)\mathbf{x_t} + c_{\text{out}}(\sigma_t)F_\theta(c_{\text{in}}(\sigma_t)\mathbf{x}_t, \sigma_t), \tag{10}$$

where $F_\theta$ is a free-form neural network, such as P-DiT, and $c_{\text{in}}$, $c_{\text{out}}$, and $c_{\text{skip}}$ control the scaling of input, output magnitudes, and the skip connection, respectively. These can be expressed as:

$$c_{\text{in}}(\sigma_t) = \frac{1}{\sqrt{\sigma_{\text{data}}^2 + \sigma_t^2}}, \quad c_{\text{skip}}(\sigma_t) = \frac{\sigma_{\text{data}}^2}{(\sigma_t - \sigma_0)^2 + \sigma_{\text{data}}^2}, \quad c_{\text{out}}(\sigma_t) = \frac{\sigma_{\text{data}}(\sigma_t - \sigma_0)}{\sqrt{\sigma_{\text{data}}^2 + \sigma_t^2}}, \tag{11}$$

These formulations satisfy the boundary condition, as $c_{\text{skip}}(\sigma_0) = 1$ and $c_{\text{out}}(\sigma_0) = 0$.

#### A.1.2  CM HYPERPARAMTERS

In the subsection, we describe the hyperparameters used for training P-DiT. Most hyperparameters for CM are adopted from iCT (Song & Dhariwal, 2024).

**Discretization Curriculum.** The discretization curriculum is designed to enhance CM training by systematically increasing the number of discretization timesteps $T$ in Eq. 1, improving the quality of generated samples. The discretization curriculum $\mathcal{C}(k)$ is defined as follows:

$$\mathcal{C}(k) = \min(s_0 2^{\frac{k}{K'}}, s_1) + 1, \quad \text{where } K' = \left\lfloor \frac{K}{\log_2 \frac{s_1}{s_0} + 1} \right\rfloor, \tag{12}$$

$k \in \{0, 1, \ldots, K\}$, and $K$ represents the total number of training iterations. $s_0$ and $s_1$ are the minimum and maximum number of discretization steps, respectively. While iCT (Song & Dhariwal, 2024) uses $s_0 = 10$ and $s_1 = 1280$, we empirically found that setting maximum number of discretization steps $s_1$ to 160 is sufficient to produce competitive performance.

**Noise Schedule.** The noise schedule plays a critical role in determining the sampling of noise levels during the CM training, significantly influencing the quality of the generated samples. For defining noise schedule, we first discretize the noise level as follows: $\sigma_{\min} = \sigma_0 < \sigma_1 < \cdots < \sigma_T = \sigma_{\max}$ where $\sigma_{\min} = 0.002$, $\sigma_{\max} = 80$. As in (Karras et al., 2022; Song et al., 2023; Song & Dhariwal, 2024), we set $\sigma_t$ as:

$$\sigma_t = \left( \sigma_{\min}^{1/\tau} + \frac{t - 1}{\mathcal{C}(k) - 1} \left( \sigma_{\max}^{1/\tau} - \sigma_{\min}^{1/\tau} \right) \right)^\tau, \tag{13}$$

where $t \in \{1, 2, \ldots, \mathcal{C}(k)\}$, and $\tau = 7$. $\tau$ controls the step length between noise levels $\sigma_t$ and $\sigma_{t+1}$. As $\tau$ increases, the step lengths at lower noise levels decrease, allowing the model to better capture high-frequency details.

Additionally, we utilize a lognormal distribution for noise level sampling, which reduces the emphasis on higher noise levels and mitigates the accumulation of errors in CT loss at lower noise levels. The noise sampling schedule is defined as:

$$\sigma_t, \text{ where } t \sim p(t), \text{ and } p(t) \propto \text{erf}\left( \frac{\log(\sigma_{t+1}) - P_{\text{mean}}}{\sqrt{2}P_{\text{std}}} \right) - \text{erf}\left( \frac{\log(\sigma_t) - P_{\text{mean}}}{\sqrt{2}P_{\text{std}}} \right), \tag{14}$$

where erf indicates error function, and $P_{\text{mean}}, P_{\text{std}}$ determine the shape of log distribution. We choose $P_{\text{mean}} = -1.1$, $P_{\text{std}} = 2.0$, following iCT (Song & Dhariwal, 2024).

**Loss Function.** In Eq. 1, we use the pseudo-Huber loss (Song & Dhariwal, 2024) as the distance function $d(\cdot)$. The pseudo-Huber loss transitions between $\mathcal{L}_1$ and $\mathcal{L}_2$ metrics and is more robust to outliers than the $\mathcal{L}_2$ metric. The pseudo-Huber loss is defined as:

$$d(\mathbf{x}, \mathbf{y}) = \sqrt{\|\mathbf{x} - \mathbf{y}\|_2^2 + c^2} - c, \tag{15}$$

where $c = 0.00054\sqrt{d}$, and $d$ indicates data dimensionality.

**Loss Weighting.** The weighting function $\lambda(\cdot)$ in Eq. 1 modulates the significance of CT losses across varying noise levels during training. As followed by (Song & Dhariwal, 2024), we set the weighting function $\lambda(\cdot)$ as follows:

$$\lambda(\sigma_t) = \frac{1}{\sigma_{t+1} - \sigma_t}. \tag{16}$$

By assigning lower weights to higher noise levels, the weighting function ensures that the model focuses on learning from lower noise levels, where the data is more distinct and informative. This strategy improves sample quality by reducing the influence of errors associated with higher noise levels, thereby enhancing the overall performance of consistency models.

### A.1.3 LATENT CODE NORMALIZATION

The EDM formulation assumes that the mean and standard deviation of the training data are zero and $\sigma_{\text{data}}$, respectively, as stated in Eq. 10. Following the approach in (Karras et al., 2024), we also normalize the encoded latent codes using precomputed statistics from the training data. Specifically, we first calculate the channel-wise mean and standard deviation of the latent codes from the training dataset. Then, we subtract the input latent code by the precomputed mean to achieve a mean of zero, and divide it by the precomputed standard deviation, followed by multiplying by $\sigma_{\text{data}}$, to set the standard deviation to $\sigma_{\text{data}}$. When the latent codes are generated, we reverse this procedure before transforming them back to the image space via the Decoder.

### A.1.4 P-DiT HYPERPARAMTERS

Following the approach in (Song et al., 2023; Song & Dhariwal, 2024), we update P-DiT parameters using an exponential moving average with a decay rate of 0.9999 to stabilize the training process. P-DiT's model hyperparameters are based on DiT-S (Peebles & Xie, 2023), except for the number of blocks ($B = 8$), to improve efficiency. The input noised latent is tokenized with a patch size of 1, allowing for finer noise information to be embedded in the latent code. To mitigate overfitting, we apply a dropout rate of 0.1 to the pointwise feed-forward layer and add minor noise and apply downsampling operation with a factor of 2 to the conditional clean images (Song et al., 2023).

### A.2 MODEL SIZE AND INFERENCE SPEED IN MFN

MFN consists of two models: PAE and P-DiT. PAE has 14.9M parameters, while P-DiT has 29.1M parameters, resulting in a total parameter count comparable to NeCA-W (Fu et al., 2023), which has 40.5M parameters across models for different camera manufacturers in the SIDD dataset. or training both PAE and P-DiT, we utilize four Nvidia A6000 GPUs. Following the approach in (Song et al., 2023; Song & Dhariwal, 2024; Karras et al., 2022; 2024), we train P-DiT using mixed-precision, which reduces both training time and memory consumption. During inference, MFN, which fully utilizes modern GPU architectures, synthesizes 57 noisy images with size of $256 \times 256$ per second, whereas NAFlow (Kim et al., 2024) generates only 13 images per second using a single A6000 GPU.

### A.3 DETAILS OF METADATA CLASSIFICATION

### A.3.1 CAMERA SENSOR AND ISO LEVEL COMBINATION

We selected camera sensors and ISO levels that overlap between SIDD training and validation sets. There are five camera sensors (*i.e.,* GP, IP, S6, N6, and G4), each paired with different combinations of ISO levels, yielding 16 different labels in total. Details regarding the labels assigned to each combination are shown in Tab. 10

### A.3.2 CLASSIFICATION MODEL ARCHITECTURE

We employed residual blocks (He et al., 2016) to construct a metadata classification model. Specifically, we sequentially use four residual blocks, each featuring a $3 \times 3$ convolutional layer with a

Table 10: List of camera sensors, ISO levels, and labels from the SIDD dataset used for metadata classification.

| Camera Sensor | ISO Level | Label | |
| --- | --- | --- | --- |
| | | *Camera Sensor* | *Camera Sensor + ISO Level* |
| GP | 1600, 3200, 6400, 10000 | 0 | 0,1,2,3 |
| IP | 1000, 1600, 2000 | 1 | 4,5,6 |
| S6 | 400, 800, 1600, 3200 | 2 | 7,8,9,10 |
| N6 | 400, 800, 3200 | 3 | 11,12,13 |
| G4 | 400, 800 | 4 | 14,15 |

stride of 2 at the front, to downsample the input prompt features, except for the last residual block. We adopt the PD operation to adjust the shape of the prompt representations from all scales to the smallest possible size that matches the spatial dimensions of the latent code $\mathbf{z}$. We then concatenate all features, aggregate them using a $1 \times 1$ convolutional layer with a stride of 2, and feed them into the classification model to produce the final output. The number of channels in the classification model is fixed at 48, resulting in a total of 778k parameters.

### A.3.3 TRAINING HYPERPARAMETERS

The classification model is trained from scratch using the Adam optimizer (Kingma & Ba, 2014) and minimize the cross-entropy loss between the predicted and ground-truth label, along with the $\mathcal{L}_2$ regularization applied to all learnable weights to address overfitting. We start with an initial learning rate 1e-4, which is then reduced to 1e-6 using a cosine annealing algorithm (Loshchilov & Hutter, 2017) over 10k iterations. For training, we use randomly cropped patches of size $256 \times 256$ and a mini-batch size of 64.

### A.4 ABLATION STUDY ON CONDITIONAL FEATURES

P-DiT generates latent codes that embed noise information through conditional features. Specifically, in Fig. 3(a), we first add the conditional feature $F_{\text{cond}}$ on the timestep embedding, and in Fig. 3(b), we condition the conditional feature through prompt attention.

In Tab. 11, we investigate the effect of conditional features on two different components: the timestep embedding and attention. P-DiT without any conditional features fails to capture the target noise information, resulting in the poorest performance in terms of KLD and AKLD. The model that conditions the conditional features on both components demonstrates superior performance compared to the model that only conditions them on the timestep embedding. This highlights that utilizing spatial information through proposed prompt attention allows the model to fully exploit the conditional features.

### A.5 EVALUATION OF THE DIVERSITY OF GENERATED NOISE

The diversity of generated images is a crucial factor in measuring the quality of a synthesized dataset. Therefore, in Tab. 12, we extend the experiment conducted in Tab. 3, which measured generated noise similarity only along the spatial axis, to one that also considers similarity along the temporal axis of continuously captured images.

To measure the KLD along the temporal axis, we use the SIDD-Full dataset, an extension of the SIDD-Medium dataset, which contains 150 noisy-clean paired images per scene captured in continuous shooting mode. We select five different scenes [1] captured by different camera sensors (*e.g.,* GP, IP, S6, N6, G4) and extract nine patches of size $256 \times 256$, uniformly spaced across each patch. We sample noise 150 times, comparing them pixel-by-pixel with the real noise, and average the KLD scores across all pixels.

As shown in Tab. 12, MFN achieves the best performance compared to the other three models in KLD scores. This indicates MFN's superior ability to sythesize noise with greater diversity.

---

[1]The indices of the selected scenes are 36, 52, 70, 99, and 169. For a more detailed description of the dataset, please refer official SIDD website.

Table 11: Effect of conditioning features on different components in P-DiT. The best results are shown in **bold**.

| Conditioning | | Generation | |
|:---:|:---:|:---:|:---:|
| Timestep | Attention | KLD↓ | AKLD↓ |
| ✗ | ✗ | 0.5661 | 0.4132 |
| ✓ | ✗ | 0.0287 | 0.1291 |
| ✓ | ✓ | **0.0261** | **0.1108** |

Table 12: Quantitative results for the diversity of synthetic noise on the SIDD-Full dataset. All methods are trained on the SIDD-Medium training set. The results are reported using KLD↓, with the best results highlighted in **bold**.

| Metric | C2N | NeCA-W | NAFlow | **MFN** |
|:---:|:---:|:---:|:---:|:---:|
| KLD↓ | 0.1248 | 0.0351 | 0.0271 | **0.0241** |

Table 13: Quantitative results on supervised denoising performance. All methods are trained with SIDD training set. *Real* refers to the original noisy-clean pairs, while MFN denotes the synthesized noisy-clean pairs generated by MFN. The percentage (%) indicates the combining ratio between the two datasets. The best and second-best results are denoted as **bold** and underline, respectively.

| *Real* / MFN | SIDD Validation | | SIDD+ | | PolyU | | Nam | | DND | | Average | |
|:---:|:---:|:---:|:---:|:---:|:---:|:---:|:---:|:---:|:---:|:---:|:---:|:---:|
| | PSNR↑ | SSIM↑ | PSNR↑ | SSIM↑ | PSNR↑ | SSIM↑ | PSNR↑ | SSIM↑ | PSNR↑ | SSIM↑ | PSNR↑ | SSIM↑ |
| 100% / 0% | 37.72 | 0.8905 | 35.68 | 0.8860 | 36.34 | 0.9204 | 35.35 | 0.8828 | 38.85 | 0.9434 | 36.79 | 0.9046 |
| 0% / 100% | 37.64 | 0.8960 | 36.23 | 0.9072 | 37.93 | 0.9609 | 38.08 | **0.9630** | 38.75 | 0.9468 | 37.73 | 0.9348 |
| 50% / 50% | **37.96** | **0.9047** | **36.57** | **0.9137** | **37.98** | **0.9610** | **38.09** | 0.9617 | **39.05** | **0.9472** | **37.93** | **0.9376** |

## A.6 QUALITATIVE RESULTS OF NOISE GENERATION

In Fig. 6, we provide additional visualizations of generated noise, comparing our method (MFN) with other approaches: C2N (Jang et al., 2021), NeCA-W (Fu et al., 2023), and NAFlow (Kim et al., 2024).

## A.7 QUALITATIVE RESULTS OF DENOISING PERFORMANCE

In Fig. 7, we provide additional visualizations of denoising results, where the denoising network is trained on synthetic datasets. We compare our method (MFN) with other approaches, including C2N, NeCA-W, and NAFlow.

## A.8 PRACTICAL APPLICATIONS OF SYNTHESIZED DATASETS FOR SUPERVISED LEARNING

In Tab. 4, 5, we demonstrate that the denoising network trained with datasets generated by the proposed MFN achieves superior performance compared to previous works and exhibits robust performance on external real-world datasets. However, for enhanced practical usage of MFN with further improvements in denoising performance, real and synthetic datasets generated by MFN can be combined in specific ratios. This approach, using the DnCNN architecture, further enhances denoising performance. As in Tab. 13, through a series of experiments, we found that adjusting the ratio of real to synthesized data and constructing a mixed dataset (50% *Real* / 50% Synthesized) leads to superior performance compared to training exclusively on the original dataset (100% *Real* / 0% Synthesized) across all evaluated real-world datasets. Specifically, this configuration maintains robustness across different camera sensor types (e.g., DSLR) on the PolyU and Nam datasets, while also outperforming the 'Real-only' (100% *Real* / 0% Synthesized) configuration on the SIDD Validation, SIDD+, and DND datasets. These results underscore that leveraging both the existing training dataset and the synthesized dataset leads to improved denoising performance, highlighting the effectiveness of our approach.

Table 14: Quantitative results on self-supervised denoising performance. All methods are trained with SIDD training set. *Real* refers to the original noisy images, while MFN denotes the synthesized noisy images generated by MFN. The percentage (%) indicates the combining ratio between the two datasets. The best and second-best results are denoted as **bold** and underline, respectively.

| *Real* / MFN | SIDD Validation | | SIDD+ | | PolyU | | Nam | | DND | | Average | |
|---|---|---|---|---|---|---|---|---|---|---|---|---|
| | PSNR↑ | SSIM↑ | PSNR↑ | SSIM↑ | PSNR↑ | SSIM↑ | PSNR↑ | SSIM↑ | PSNR↑ | SSIM↑ | PSNR↑ | SSIM↑ |
| 100% / 0% | 35.99 | 0.8630 | 35.23 | 0.8975 | 37.01 | 0.9517 | 37.19 | 0.9566 | 37.36 | 0.9266 | 36.56 | 0.9191 |
| 0% / 100% | 35.71 | 0.8630 | **35.76** | **0.9164** | 37.31 | 0.9542 | 37.42 | 0.9570 | 37.35 | 0.9250 | 36.71 | 0.9231 |
| 50% / 50% | **36.94** | **0.9019** | 35.56 | 0.9152 | **37.44** | **0.9570** | **37.76** | **0.9621** | **38.29** | **0.9412** | **37.20** | **0.9355** |

Table 15: Quantitative results on denoising generalization performance depending on the size of synthesized dataset using MFN. The multiplication sign (×) indicates the scaling applied to the original number of patches. The best and second-best results are denoted as **bold** and underline, respectively.

| # Samples | PolyU | | Nam | | SIDD Validation | | SIDD+ | | Average | |
|---|---|---|---|---|---|---|---|---|---|---|
| | PSNR↑ | SSIM↑ | PSNR↑ | SSIM↑ | PSNR↑ | SSIM↑ | PSNR↑ | SSIM↑ | PSNR↑ | SSIM↑ |
| ×1 | 37.40 | **0.9569** | 37.29 | 0.9578 | 36.55 | 0.8887 | 35.72 | 0.8997 | 36.74 | 0.9258 |
| ×2 | 37.61 | 0.9546 | 37.92 | **0.9580** | 36.95 | 0.8838 | 35.87 | **0.9078** | 37.09 | 0.9260 |
| ×4 | **37.72** | 0.9542 | **37.94** | 0.9578 | **37.27** | **0.8955** | **36.00** | 0.9022 | **37.23** | **0.9274** |

## A.9 REAL-WORLD SELF-SUPERVISED DENOISING VIA NOISE GENERATION

We further evaluated the efficacy of the proposed MFN on self-supervised denoising methods. As in Tab. 14, we trained LGBPN (Wang et al., 2023) using noisy images from three different combinations of real SIDD data (100% *Real* / 0% Synthesized), fully synthesized data (0% *Real* / 100% Synthesized), and mixed data (50% *Real* / 50% Synthesized). The results exhibit a similar trend to those in Tab. 13, where incorporating synthetic data into the training process enhances performance across various real-world datasets. Specifically, using a mixed dataset (50% *Real* / 50% Synthesized) significantly boosts performance across all real-world datasets compared to the 'Real-only' (100% *Real* / 0% Synthesized) configuration, yielding the best averaged results. These findings underscore that leveraging both the existing training dataset and the synthesized dataset leads to improved denoising performance, demonstrating the effectiveness of our approach in both supervised and self-supervised learning contexts.

## A.10 FURTHER ANALYSIS OF THE GENERALIZATION PERFORMANCE

We also conducted experiments to analyze the robustness of our method on external datasets. Our results show that expanding the dataset size through synthetic image generation helps mitigate overfitting to the training distribution, leading to improved performance on external datasets.

For the experimental setup, we selected 15,000 non-overlapping patches from the SIDD training data to generate synthetic noisy images and train the DnCNN denoising network. This approach ensures the creation of unique noisy samples without repetition, providing a controlled environment for systematically analyzing the factors influencing performance on external datasets.

As illustrated in Tab. 15, we organized the training data into three distinct groups: ×1, ×2, and ×4. These factors represent the multiplicative scaling applied to the original number of patches for generating synthetic noisy images using MFN for each clean image. Importantly, performance on the in-distribution dataset (SIDD+) remains consistent across all groups. The denoising network shows enhanced robustness across all real-world datasets as the number of samples increases, with the ×4 group achieving the best average results. These findings suggest that leveraging multiple noise samples effectively reduces overfitting by increasing the diversity of the noise, thereby underscoring the effectiveness of the MFN approach.

## A.11 EVALUATION ON REAL-WORLD MEDICAL IMAGING DATASET

Several previous studies (Abdelhamed et al., 2019; Kousha et al., 2022; Fu et al., 2023) utilize metadata to guide the modeling of specific noise types. However, in practical applications such as medical imaging, the physical meaning of the metadata may differ entirely or may even be un-

Table 16: Quantitative results of noise generation performance on LDCT images in terms of KLD↓ and AKLD↓. Note that the best results are highlighted in **bold**.

| Methods | KLD↓ | AKLD↓ |
|---------|------|-------|
| NAFlow | 0.0555 | 0.2776 |
| Ours | **0.0517** | **0.1433** |

Table 17: Quantitative results of denoising performance on LDCT images in terms of PSNR↑ and SSIM↑. The percentage (%) indicates the combining ratio between the two datasets. Note that the best results are highlighted in **bold**.

| Methods | PSNR↑ | SSIM↑ |
|---------|-------|-------|
| *Real* | 44.57 | 0.9612 |
| NAFlow | 43.79 | 0.9559 |
| Ours | 44.20 | 0.9583 |
| *Real* + Ours (50% / 50%) | **44.70** | **0.9617** |

available. Consequently, conventional methods that rely on standardized metadata face significant limitations.

To address this limitation and demonstrate the generalizability of our proposed MFN to other imaging domains where metadata is unavailable or has different physical meanings, we trained MFN using a medical imaging dataset. Specifically, we utilized a real-world low-dose (LD) CT image dataset (McCollough et al., 2017), which includes 1mm and 3mm-thick abdominal slices with B30 and D45 kernels. This dataset comprises noisy quarter-dose images and their corresponding ground-truth normal-dose (ND) images. For the training and test datasets, we select 15,154 paired images from nine patients and 1,474 images from patient L506, respectively. Unlike sRGB images, where noise characteristics are influenced by factors such as ISO settings and camera sensor types, noise in CT images is affected by a range of factors, including X-ray dose, slice thickness, and reconstruction algorithms.

To evaluate the noise modeling performance of MFN, we compare the results with those of the recent state-of-the-art model, NAFlow (Kim et al., 2024). While NAFlow incorporates metadata, such as slice thickness and reconstruction algorithms, during training, MFN is trained and tested without any metadata or prior knowledge of CT imaging, demonstrating its strength in real-world applications. As shown in Tab. 16, MFN significantly outperforms NAFlow in noise quality metrics, including both KLD and AKLD.

In Tab. 17, we further report the denoising performance using the NAFNet (Chen et al., 2022) architecture. When trained with MFN, NAFNet achieves superior results in terms of PSNR and SSIM compared to NAFlow. Furthermore, when trained with a mixed dataset comprising 50% original and 50% synthesized data, MFN outperforms the denoising network trained exclusively on the *Real* dataset. These results align with the findings presented in Tab. 13 and Tab. 14, indicating that the MFN effectively generalizes to medical imaging contexts, such as CT scans, even in the absence of metadata. This underscores the potential of the proposed MFN for practical applications across various imaging domains, further highlighting its promising capabilities.

### A.12 MOTIVATION FOR ADOPTING TWO-STAGE TRAINING SCHEME

Our framework comprises PAE, an autoencoder designed for latent code embedding and noisy image reconstruction, and P-DiT, a consistency model for latent code generation. During inference, the generated latent code is input into the decoder to produce the final synthesized noisy image. Consequently, both reconstruction performance and generative capabilities are essential for generating high-quality noisy images. However, as highlighted in (Rombach et al., 2022b), a joint training pipeline requires a careful balance between reconstruction and generative performance. This necessitates extensive experimentation to determine the optimal weight, a process that is both time- and resource-intensive. To address this challenge, we propose a two-stage framework, MFN, in which the reconstruction network is initially trained and then frozen while the consistency model is trained, thereby ensuring training stability.

Table 18: Quantitative results for the noise generation performance on SIDD validation set under both paired and unpaired settings. The results are reported using KLD↓ and AKLD↓, with the best and second-best results highlighted in **bold** and underline, respectively.

| Model | Given $\mathcal{I}_{\text{Clean}}^{\text{Test}}$ at test phase? | KLD↓ | AKLD↓ |
|---|---|---|---|
| NeCA-W | ✗ | 0.0342 | 0.1436 |
| NAFlow | ✓ | 0.0305 | 0.1306 |
| MFN (Unpaired) | ✗ | 0.0226 | 0.1223 |
| MFN (Paired) | ✓ | **0.0194** | **0.1163** |

## A.13 EVALUATION OF UNPAIRED NOISE GENERATION

To generate the final noisy image $\hat{\mathcal{I}}_{\text{Noisy}}$, we use paired noisy image $\mathcal{I}_{\text{Noisy}}$ and clean image $\mathcal{I}_{\text{Clean}}$ during both the training and inference phases to compute $\mathbf{n}_{\text{Real}} = \mathcal{I}_{\text{Clean}} - \mathcal{I}_{\text{Noisy}}$. This paired strategy ensures optimal performance. However, to further extend the application of the proposed MFN to scenarios where only clean images $\mathcal{I}_{\text{Clean}}$ are available, we evaluated MFN in an unpaired setting. In this scenario, noise from the training dataset $\mathbf{n}_{\text{Real}}^{\text{Meta}}$, which is sampled based on the target metadata settings (e.g., camera, ISO), is used to synthesize noisy images with an unpaired clean image $\mathcal{I}_{\text{Clean}}^{\text{Test}}$ during the test phase.

As summarized in Tab. 18, the results on the SIDD validation set show that while the unpaired setting exhibits a slight performance degradation compared to the paired setting, MFN still achieves comparable or superior noise quality metrics, in terms of KLD and AKLD, when compared to other SOTA methods, such as NeCA-W and NAFlow. This flexibility underscores the adaptability of MFN, as our framework effectively synthesizes noisy images without the need for paired noisy-clean images. This makes it particularly suitable for scenarios where paired datasets are unavailable. Additionally, in the paired setting, MFN demonstrates its ability to handle noise effectively without the need for explicitly defined metadata, such as ISO or camera type. These results further reinforce the practicality of MFN across a wide range of use cases without requiring any modifications, regardless of whether paired or unpaired data is available.

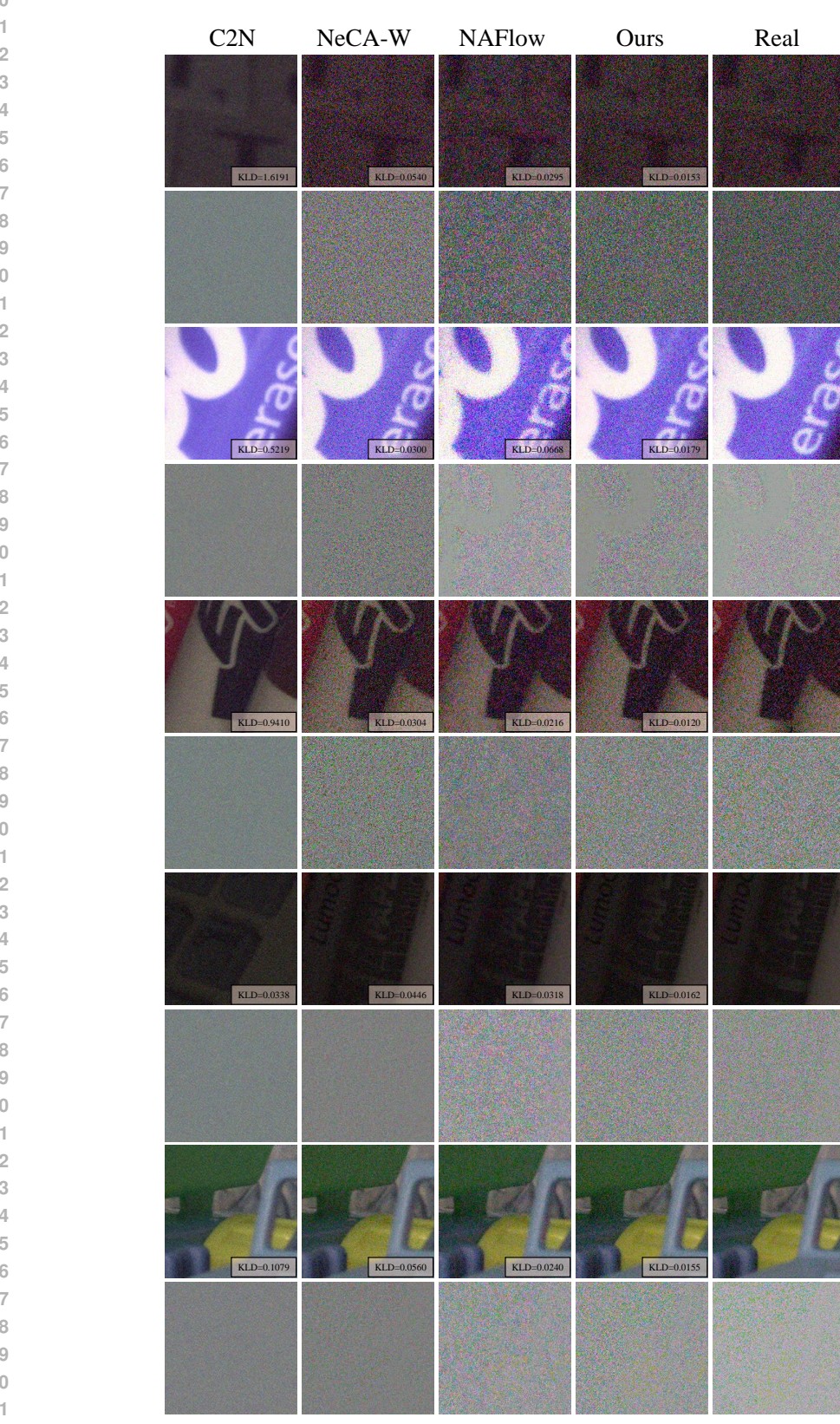

Figure 6: Visualization of synthetic noisy images on the SIDD validation set. From left to right: C2N, NeCA-W, NAFlow, Ours (MFN), and real noisy images.

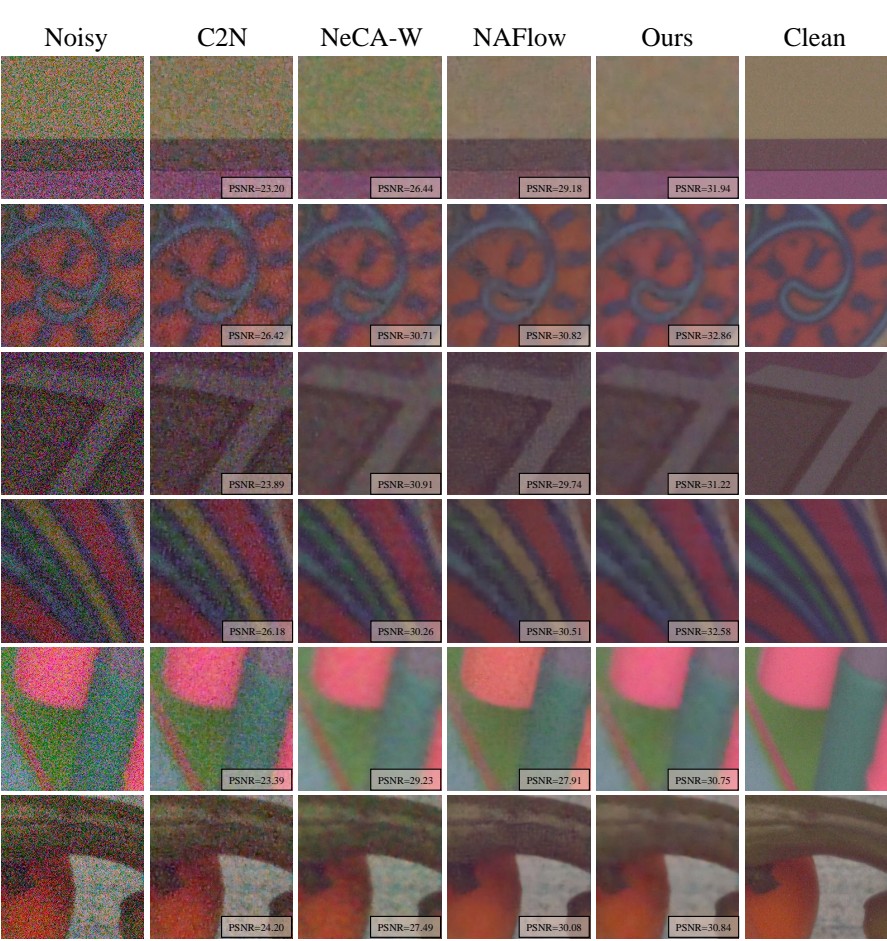

Figure 7: Visual comparison on denoising results with PSNR↑ on SIDD validation set from DnCNN trained on each method.

