# OpenReview forum: "MFN: Metadata-Free Real-World Noisy Image Generation"
_ICLR.cc/2025/Conference — Submitted to ICLR 2025_

### Official Review · Reviewer_kgBq · 2024-11-02

**Soundness:** 1
**Presentation:** 4
**Contribution:** 2
**Rating:** 3
**Confidence:** 5

**Summary:**

The paper presents a method for generating real-world noisy images without relying on meta information, which seems to be useful in medical imaging. It consists a Prompt Autoencoder to encodes noise characteristics into a latent space. The Prompt Diffusion Transformer is utilized to integrate the prompt features from PAE  and generate the noisy images. The experiments are conducted on several datasets.

**Strengths:**

1. The method operates without the need for metadata, which could be especially beneficial for certain specialized sensors and applications
2. The paper is well-written..

**Weaknesses:**

1. During inference, the method requires real noise as input. In practice, we could directly use noise and clean images to create noisy-clean image pairs, which would be ideal for training and work better that the proposed method.
2. Although the paper claims (L49) that the metadata-free setting is reasonable, it does not provide real-world application examples, such as in medical imaging. It remains unclear whether the method can generalize to such contexts.
3. The model was trained on the SIDD dataset but performs well on other datasets, even when noise characteristics differ significantly, which is somewhat unexpected.
4. More discussion should be conduct with the existing physics-based noise modeling paper [1-2].

[1] A Physics-based Noise Formation Model for Extreme Low-light Raw Denoising

[2] Rethinking noise synthesis and modeling in raw denoising

**Questions:**

See the strengths and weaknesses

---

> ### Author Response · Authors · 2024-11-22
>
> Dear Reviewer kgBq,
>
> We would like to thank the reviewer for taking time to review our manuscript and acknowledging our contributions. Below is our response to reviewer’s concerns.
>
> ---
>
> > **(W1) Motivation of Synthesizing Real-World Dataset**
>
> We agree that directly using collected real-world noisy-clean image pairs for the supervised training of a denoising network offers certain advantages. However, gathering a large-scale real-world dataset is highly labor-intensive, time-consuming, and costly. Additionally, the limited size and diversity of such datasets often lead to overfitting due to the constrained variance in their distributions.
>
> To address these challenges, we propose a dataset synthesis approach based on the MFN framework. This method significantly reduces the reliance on extensive real-world data collection while maintaining practical effectiveness.
>
> In Table 4 of the main manuscript, we observe that training the denoising network exclusively on the synthesized dataset leads to slightly degraded performance on the SIDD dataset. This can be attributed to the complex nature of sRGB noise, a challenge that has been addressed in previous works [1*, 2*, 3*, 4*], which also faced difficulties surpassing the performance of models trained on real data.
>
> However, to make the proposed MFN model more effective for practical applications, combining real and synthetic datasets generated by MFN can significantly enhance denoising performance. By adjusting the ratio of real to synthesized data, we found that a mixed dataset (50% Real / 50% Synthesized) achieves superior performance compared to training solely on the original dataset (100% Real / 0% Synthesized) across all evaluated datasets. Specifically, this configuration maintains robustness across different camera sensor types (e.g., DSLR) on the PolyU and Nam datasets, while also outperforming the 'Real-only' (100% Real / 0% Synthesized) configuration on the SIDD Validation, SIDD+, and DND datasets.
>
> These results underscore leveraging both the existing training dataset and the synthesized dataset leads to improved denoising performance, showcasing the effectiveness of our approach.
>
> | MFN + DnCNN | SIDD Validation | SIDD+ | PolyU | Nam | DND | Average |
> | --- | --- | --- | --- | --- | --- | --- |
> | Real / Synthesized (%) | PSNR↑ / SSIM↑ | PSNR↑ / SSIM↑ | PSNR↑ / SSIM↑ | PSNR↑ / SSIM↑ | PSNR↑ / SSIM↑ | PSNR↑ / SSIM↑ |
> | 100 / 0 ($Real$ in Tab. 5) | 37.72 / 0.8905 | 35.68 / 0.8860 | 36.34 / 0.9204 | 35.35 / 0.8828 | 38.85 / 0.9434 | 36.79 / 0.9046 |
> | 0 / 100 (Ours in Tab. 5) | 37.64 / 0.8960 | 36.23 / 0.9072 | 37.93 / 0.9609 | 38.08 / **0.9630** | 38.75 / 0.9468 | 37.73 / 0.9348 |
> | 50 / 50 | **37.96** / **0.9047** | **36.57** / **0.9137** | **37.98** / **0.9610** | **38.09** / 0.9617 | **39.05** / **0.9472** | **37.93** / **0.9376** |
>
> ---
>
> > **(W2) Real-World Medical Image Denoising.**
>
> Thank you for the valuable suggestion.
>
> To evaluate the robust performance of the proposed MFN framework in other imaging domains, such as medical imaging, we selected the AAPM-Mayo Clinic Low Dose CT Grand Challenge dataset [5*], which consists of noisy-clean paired CT images. Unlike sRGB images, where noise characteristics are influenced by factors like ISO and camera sensor types, noise in CT images is affected by other various factors, including X-ray dose, slice thickness, and reconstruction algorithms.
>
> We evaluated the noise modeling and denoising performance of MFN compared to NAFlow. While NAFlow utilizes metadata, such as slice thickness and reconstruction algorithms, during training, MFN is trained and tested without any metadata or prior knowledge about CT imaging. The results show that MFN outperforms NAFlow in noise quality metrics, including KLD and AKLD, as shown in the table below.
>
> |  | KLD**↓** | AKLD**↓** |
> | --- | --- | --- |
> | NAFlow | 0.0555 | 0.2776 |
> | **MFN** | **0.0517** | **0.1433** |
>
> For denoising performance, as shown in the table below, a denoising network using the NAFNet [6*] architecture trained with MFN achieves superior results in terms of PSNR and SSIM compared to NAFlow. Furthermore, when trained with a mixed dataset comprising 50% original and 50% synthesized data, MFN outperforms the denoising network trained exclusively on the $Real$ dataset.
>
> These results are consistent with the findings in the above (W1) and (W2) sections from Reviewer GejD, demonstrating that MFN generalizes effectively to medical imaging contexts, such as CT, even without the use of metadata. This highlights the potential of the proposed MFN for practical applications across diverse imaging domains, emphasizing its promise.
>
> |  | PSNR↑ | SSIM↑ |
> | --- | --- | --- |
> | NAFNet + $Real$ | 44.57 | 0.9612 |
> | NAFNet + NAFlow | 43.79 | 0.9559 |
> | NAFNet + MFN | 44.20 | 0.9583 |
> | **NAFNet + $Real$ + MFN (50% / 50%)** | **44.70** | **0.9617** |

---

> ### Author Response · Authors · 2024-11-22
>
> > **(W3) Further Analysis of the Generalization Performance**
>
> As discussed in the (W1) section, our primary motivation for modeling noise distributions is to alleviate the overfitting problem caused by limited dataset size through dataset synthesis. Empirically, we demonstrate that both supervised (Section [W1]) and self-supervised (Section [W4] of Reviewer GejD) networks exhibit robustness on external benchmark datasets when trained with our synthesized dataset, either solely or in a mixed manner.
>
> To further analyze the factors that impact robustness on external datasets, we conducted an additional experiment. Our findings indicate that increasing the dataset size by generating numerous noise samples with MFN effectively reduces overfitting to the training distribution and enhances performance on external datasets.
>
> We used 15,000 non-overlapping patches from the SIDD training data to generate synthetic noisy images and train the DnCNN denoising network. This approach ensures unique noisy samples without repetition and creates a controlled environment to systematically analyze the factors influencing performance on external datasets.
>
> As shown in the table below, we created three separate training groups: $\times$1, $\times$2, and $\times$4. These factors represent the multiplication applied to the original number of patches to generate synthetic noisy images using MFN per clean image. Note that the performance on the in-distribution dataset (SIDD+) remains consistent. The denoising network demonstrates improved robustness across all real-world datasets as the number of samples increases, with $\times$4 producing the best average results. These findings suggest that utilizing multiple noise samples effectively reduces overfitting, highlighting the effectiveness of MFN.
>
> | DnCNN + MFN | PolyU | Nam | SIDD Validation | SIDD+ | Average |
> | --- | --- | --- | --- | --- | --- |
> | # Samples | PSNR↑ / SSIM↑ | PSNR↑ / SSIM↑ | PSNR↑ / SSIM↑ | PSNR↑ / SSIM↑ | PSNR↑ / SSIM↑ |
> | $\times$1 | 37.40/**0.9569** | 37.29/0.9578 | 36.55/0.8887 | 35.72/0.8997 | 36.74/0.9258 |
> | $\times$2 | 37.61/0.9546 | 37.92/**0.9580** | 36.95/0.8838 | 35.87/**0.9078**  | 37.09/0.9260  |
> | $\times$4 | **37.72**/0.9542 | **37.94**/0.9578  | **37.27**/**0.8955** | **36.00**/0.9022 | **37.23**/**0.9274**  |
>
> ---
>
> > **(W4) Comparsion with Physics-based Noise Modeling.**
>
> Thank you for suggesting the papers for a more in-depth discussion. Physics-based noise modeling approaches, as presented in [7*] and [8*], approximate noise distribution using physical equations derived from the characteristics of CMOS photosensors. These models are particularly suited for the **raw-RGB** domain, where noise is predominantly signal-dependent and less influenced by non-linear processing.
>
> In contrast, our work focuses on noise modeling in the **sRGB** space, which is widely used for web content and general-purpose displays. Noise modeling in sRGB presents additional challenges compared to the raw-RGB space. In sRGB images, noise is heavily altered by the image signal processor (ISP) pipeline, which includes non-linear operations such as demosaicing, white balancing, and tone mapping. These processes not only amplify the original sensor noise but also introduce complex, content-dependent artifacts. Consequently, modeling noise in the sRGB space requires accounting for these intricate distortions. For further details, please refer to Figure 2 in [2*, 9*].
>
> To address these complexities, we propose using Global and Local Prompt Blocks. The Global Prompt Block captures signal gain-related characteristics, while the Local Prompt Block models spatially complex, non-linear patterns introduced by the ISP pipeline. Together, these components enable our framework to encode the challenging distortions of sRGB noise into the latent space. For further details, please refer to the W2 section of Reviewer GejD.
>
> While physics-based approaches provide a solid foundation for understanding noise formation, our work focuses on the intricate challenges of sRGB noise modeling and presents an effective approach for addressing these complexities.
>
> ---
>
> > **Reference**
>
> [1*] C2N: Practical generative noise modeling for real-world denoising, ICCV 2021
>
> [2*] Modeling sRGB Camera Noise with Normalizing Flows, CVPR 2022
>
> [3*] sRGB Real Noise Synthesizing with Neighboring Correlation-Aware Noise Model, CVPR 2023
>
> [4*] sRGB Real Noise Modeling via Noise-Aware Sampling with Normalizing Flows, ICLR 2024
>
> [5*] Low-dose CT for the detection and classification of metastatic liver lesions: Results of the 2016 Low Dose CT Grand Challenge, Medical Physics 2016
>
> [6*] Simple baselines for image restoration, ECCV 2022
>
> [7*] A Physics-based Noise Formation Model for Extreme Low-light Raw Denoising, CVPR 2020
>
> [8*] Rethinking noise synthesis and modeling in raw denoising, ICCV 2021
>
> [9*] AP-BSN: Self-Supervised Denoising for Real-World Images via Asymmetric PD and Blind-Spot Network, CVPR 2022

---

> ### Author Response · Authors · 2024-11-22
>
> Thank you for your valuable feedback and patience. We are pleased to inform you that the (W3) experiment is now complete, and we have updated the comment with the results. We look forward to any further feedback you may have.

---

> ### Author Response · Authors · 2024-11-25
>
> We appreciate your thoughtful reviews and comments.
>
> We hope our responses address your concerns and provide detailed responses. If you have any further questions or need additional clarification on any aspect, please do not hesitate to let us know.
>
> Once again, we appreciate your time and insights.

---

> ### Author Response · Authors · 2024-12-02
>
> Dear Reviewer kgBq,
>
> Thank you for your valuable contribution to the development of our paper.
>
> As the discussion period draws to a close, we kindly look forward to your response.
>
> Please let us know if you have any remaining concerns or if there is anything further we can clarify to assist you.

---

### Official Review · Reviewer_ncU9 · 2024-11-03

**Soundness:** 3
**Presentation:** 3
**Contribution:** 3
**Rating:** 8
**Confidence:** 3

**Summary:**

The paper presents a Metadata-Free Noise (MFN) model, designed to generate realistic noisy images without relying on explicit camera metadata. This framework is particularly useful in practical applications where metadata might be unavailable, such as in medical imaging or low-level vision tasks. MFN uses a Prompt Autoencoder (PAE) and Prompt DiT (P-DiT) components to capture noise characteristics, allowing for the generation of noise that resembles real-world conditions. The model’s performance is demonstrated through extensive experiments on various datasets, showcasing its effectiveness in generating realistic noise and achieving strong results in denoising applications.

**Strengths:**

Contribution: MFN’s ability to generate realistic noise without requiring metadata significantly broadens its application scope, especially in fields where metadata is unavailable or irrelevant, addressing a notable gap in real-world denoising research.

Method: The dual-component framework (PAE and P-DiT) is well-conceived. By leveraging prompt features instead of metadata, the model effectively captures and recreates noise characteristics, enhancing its realism and robustness.

Experiments: The paper presents comprehensive evaluations across multiple datasets (SIDD, PolyU, NAM, and DND) and denoising tasks. The results consistently demonstrate MFN’s superior performance in noise synthesis and denoising.

Comparison with SOTA: MFN outperforms various existing models (e.g., NAFlow, NeCA-W) on key metrics like Kullback-Leibler Divergence (KLD) and PSNR, showcasing both quantitative and qualitative improvements in noise generation and denoising quality.

Ablation studies: The authors include thorough ablation studies that highlight the contribution of different model components, such as the Global Prompt Block (GPB) and Local Prompt Block (LPB), to the quality of the generated noise.

**Weaknesses:**

Computational complexity: While MFN demonstrates strong performance, its two-stage architecture may be computationally intensive, particularly for high-resolution images. The paper could benefit from discussing the model’s runtime and potential optimizations for real-time applications.

Limited comparison: Although MFN is compared with several state-of-the-art methods, the selection could be expanded to include other prompt-based and noise-aware generation models, which would provide a more robust evaluation of its unique contributions.

Potential overfitting: The model’s performance on specific datasets such as SIDD+ is noted, but additional evaluations in extreme or highly diverse noise scenarios could further validate its generalization.

Dependence on training setup: MFN’s performance depends heavily on the training setup, which is highly specialized (e.g., custom noise schedules and curriculum training). This setup may limit reproducibility and adaptability in different research settings.

**Questions:**

What could be potential solutions to enhance the method's efficiency, especially for real time applications? How efficient is the proposed method in terms of higher image resolutions?

---

> ### Author Response · Authors · 2024-11-21
>
> Dear Reviewer ncU9,
>
> We would like to thank the reviewer for recognizing our contribution and pointing out the specific strength of our method, as well as constructive concerns. We address the reviewer's concerns as follows.
>
> ---
>
> > **(W1) Computational Complexity**
>
> To evaluate the computational complexity of our two-stage framework, MFN, we compared the inference time at different image resolutions with two other SOTA models, NeCA-W and NAFlow. Specifically, we measured the number of images that each model can process per second on a single A6000 GPU.
>
> With a resolution of 256$\times$256, MFN can generate 57 images per second, which is about 4.38 times faster than NAFlow. At a 512$\times$512 resolution, MFN generates 21 images per second, 2.625 times faster than NAFlow. At the highest resolution of 1024$\times$1024, MFN still performs 1.25 times faster than NAFlow, producing 5 images per second, demonstrating its practicality at high image resolutions.
>
> Even though NeCA-W shows the highest speed at all resolutions, the inference time difference ratio between NeCA-W and MFN decreases as the resolution increases, from 2.63 to 1.8. Hence, our method demonstrates strong performance while maintaining reasonable computational complexity at high resolutions compared to recent SOTA models.
>
> | Inference Time (images / second) | NeCA-W | NAFlow | MFN |
> | --- | --- | --- | --- |
> | 256$\times$256 | 150  | 13  | **57**  |
> | 512$\times$512 | 38  | 8  | **21**  |
> | 1024$\times$1024 | 9  | 4  | **5**  |
>
> ---
>
> > **(W2) Limited Comparison**
>
> In our work, we compared the proposed MFN with state-of-the-art noise generation methods, including NeCA  and NAFlow. Following the reviewer's recommendation, we expanded our comparisons to include additional methods such as DANet [1*], CycleISP [2*], and PNGAN [3*].
>
> The table below presents the noise generation performance of these methods measured using the AKLD metric on SIDD validation dataset. Notably, MFN achieves the lowest AKLD score, demonstrating superior performance in modeling noise distributions.
>
> | Method | DANet | GDANet | CycleISP | C2N | PNGAN | NeCA-W | NAFlow | **MFN** |
> | --- | --- | --- | --- | --- | --- | --- | --- | --- |
> | AKLD ↓ | 0.212 | 0.253 | 0.716 | 0.213 | 0.153 | 0.144 | 0.131 | **0.116** |
>
> While NAFlow and PNGAN are notable for their noise-aware methodologies, our proposed MFN further advances the field by embedding noise-relative features into prompt features, achieving state-of-the-art performance. This expanded comparison strengthens the evaluation of MFN's unique contributions and highlights its robust performance across diverse real-world benchmarks.
>
> ---
>
> > **(W3) Potential Overfitting**
>
> In Tables 2, 3, 4 and 5 in the main manuscript, we evaluate the generalization capabilities of the proposed method using three additional datasets: SIDD+, PolyU, and Nam. These datasets present diverse and challenging noise scenarios that differ from the original SIDD dataset. SIDD+ is collected using additional smartphone models which are not included in the SIDD dataset and this dataset features extreme noise levels due to the sensor limitations of these devices. This provides a valuable test for the model's ability to handle more severe noise conditions. The PolyU and Nam datasets, on the other hand, are captured using DSLR cameras. These devices have fundamentally different camera structures, sensors, and ISP pipelines compared to smartphones, resulting in distinct noise patterns and distributions. Evaluation on these datasets demonstrates the model's ability to generalize across various devices with vastly different noise characteristics.
>
> We acknowledge the importance of further validating the model's robustness in even more diverse real-world noise scenarios. We look forward to conducting additional evaluations as more benchmark datasets become available.
>
> ---
>
> > **(W4) Dependence on Training Setup**
>
> The training setup of MFN, including noise schedules and curriculum training, primarily adopts methodologies established in prior works such as iCT [4*] and EDM [5*]. To ensure reproducibility, we provide detailed hyperparameter settings and implementation specifics in Appendix A.1 of the main manuscript. These include the CM parameterization, CM hyperparameters, and P-DiT hyperparameters. Additionally, we plan to release the code upon acceptance or can be immediately opened if required by reviewers.

---

> ### Author Response · Authors · 2024-11-21
>
> > **(Q1) Potential Solutions for Enhancing Efficiency**
>
> To enhance the efficiency of MFN, our framework is designed with two key components: an autoencoder and a consistency model. The autoencoder compresses the image into a compact latent space, significantly reducing the computational cost of attention operations within the transformer-based generative models. Additionally, we employ a consistency model, which requires only a single generation step, achieving efficiency while maintaining high-quality outputs compared to traditional diffusion models.
>
> Regarding higher-resolution images, the latent space representation and single-step generation process already contribute to computational efficiency. However, scaling to very high resolutions may benefit from techniques such as model quantization [6*, 7*] or pruning [8*], which can further optimize processing time while preserving image quality. These strategies could help adapt the MFN for real-time, high-resolution scenarios.
>
> ---
>
> > **Reference**
>
> [1*] Dual adversarial network: Toward real-world noise removal and noise generation, ECCV 2020
>
> [2*] Cycleisp: Real image restoration via improved data synthesis, CVPR 2020
>
> [3*] Learning to Generate Realistic Noisy Images via Pixel-level Noise-aware Adversarial Training, NeurIPS 2021
>
> [4*] Improved Techniques for Training Consistency Models, ICLR 2024
>
> [5*] Elucidating the Design Space of Diffusion-Based Generative Models, NeurIPS 2022
>
> [6*] Q-diffusion: Quantizing diffusion models, ICCV 2023
>
> [7*] Post-Training Quantization on Diffusion Models, CVPR 2023
>
> [8*] Structural Pruning for Diffusion Models, NeurIPS 2023

---

### Official Review · Reviewer_yT2B · 2024-11-04

**Soundness:** 4
**Presentation:** 3
**Contribution:** 2
**Rating:** 8
**Confidence:** 4

**Summary:**

This paper presents a neural network for generating images with realistic camera noise from noise-free images without metadata. The proposed method incorporates a diffusion model, two-stage training, and transformer-based global and local attention. Experimental results present that the proposed method outperforms the compared methods in noise generation, image denoising, and metadata classification.

**Strengths:**

The proposed method is reasonable, and the ablation studies present the effectiveness of each technique component.

The performance improvements on various real-world denoising datasets (Table 5) indicate a practical use case of noise generation using noisy-clean image pairs.

The experiments on metadata classification are interesting for evaluating the performance of noise generation.

**Weaknesses:**

The literature review in the introduction and related work is limited.
For real-world image denoising, the works [A, B, C] have addressed different approaches worth discussing and comparing in experiments (if possible) for future research.
In particular, NERDS generates noisy-clean image pairs without metadata and clean images using raw-sRGB noisy image pairs.

The analysis for the generalization performance on various real-world datasets (Table 5) is limited.
Given that the proposed method uses noisy-clean image pairs to train noise generators while MFN and ‘real’ present similar performances in the SIDD benchmark, the generalization performance is the major advantage of using the proposed method for real-world image denoising.
However, the manuscript does not incorporate any further analysis of the generalization performance.
For instance, the restoration performance comparisons on different numbers of noisy images per clean image and the noise diversity control can help us better understand generalizations about real-world denoising.


[A] Fbi-denoiser: Fast blind image denoiser for poisson-gaussian noise, CVPR 2021

[B] Practical blind denoising via swin-conv-unet and data synthesis, MIR 2023

[C] NERDS: A General Framework to Train Camera Denoisers from Raw-RGB Noisy Image Pairs, ICLR 2023

**Questions:**

Why should the training stages be separated? And how does the separation affect performances?

---

> ### Author Response · Authors · 2024-11-22
>
> Dear Reviewer yT2B,
>
> We would like to thank the reviewer for recognizing our contribution and pointing out the specific strength of our method, as well as constructive concerns. We address the reviewer's concerns as follows.
>
> ---
>
> > **(W1) Comparision with [A,B,C]**
>
> We appreciate the reviewer for recommending [A, B, C] for comparison. Below are our responses to each of the referenced papers.
>
> **[A]** This paper primarily focuses on self-supervised image denoising, where the PGE-Net is used to estimate parameters of Poisson-Gaussian noise. In contrast, our MFN focuses on metadata-free noise generation and uses synthesized noisy-clean image pairs as a data augmentation method to improve the generalization ability of denoising networks trained in a supervised manner. For this reason, a direct comparison between this method and our MFN is not possible. Regarding generalization in a self-supervised manner, we conducted experiments on the LGBPN model in Section W4 of Reviewer GejD, where using mixed data (50% $Real$ / 50% synthesized) significantly improved denoising robustness across different real-world datasets.
>
> **[B]** We attempted to follow the proposed noise generation pipeline to create the dataset; however, the authors did not release official code for generating synthetic noisy-clean image pairs. Furthermore, the denoising model architecture and the training data used for real-world denoising differ from ours, and the authors did not report PSNR or SSIM scores for real-world denoising. For these reasons, this method cannot be directly compared with our MFN.
>
> **[C]** This work solely utilizes raw-RGB images to generate paired synthesized noisy-clean images by estimating Poisson-Gaussian noise and applying a downscaling operation, whereas our MFN uses paired sRGB noisy-clean images to generate multiple noisy images. We acknowledge the strength of NERDS, as it does not require clean signals when generating noisy images or training a denoising network.
>
> The table below summarizes the main differences between NERDS and the proposed MFN, and compares their denoising performance on the SIDD and DND benchmark datasets. Since NERDS imposes fewer constraints by not requiring clean images, the DnCNN trained with NERDS demonstrates inferior performance compared to MFN on both datasets.
>
> On the other hand, since MFN does not require raw-RGB images, our method has an advantage in real-world applications where raw-RGB images may not be available. Additionally, we have also evaluated its generation capability on unpaired data in Section (W3-ii) of Reviewer GejD, where our method still outperforms NeCA-W and NAFlow, the SOTA models, even in the unpaired setting.
>
> |  | Requires RAW-sRGB Noisy images? | Requires Clean images? | SIDD Benchmark (PSNR↑/SSIM↑) | DND (PSNR↑/SSIM↑) |
> | --- | --- | --- | --- | --- |
> | DnCNN+NERDS | o | x |  36.42/0.923 | 38.21/0.941 |
> | **DnCNN+MFN** | x | o |  **37.63**/**0.936** | **38.75**/**0.947** |

---

> ### Author Response · Authors · 2024-11-22
>
> > **(W2) Further Analysis of the Generalization Performance**
>
> We thank the reviewers for recognizing the robust performance of our proposed method on external datasets. Additionally, we would like to highlight that for more practical applications of MFN, combining real and synthesized datasets enables both supervised and self-supervised denoising networks to improve performance on the SIDD Validation dataset while maintaining strong performace to external datasets such as SIDD+, PolyU, Nam, and DND (Sections [W1] and [W4] of Reviewer GejD). Furthermore, we extend our MFN framework to synthesize noise in medical imaging, such as CT (Section [W2] of Reviewer kgBq).
>
> As suggested by the reviewer, we conducted experiments to analyze the robustness of our method on external datasets. Our findings indicate that increasing the dataset size through generation helps alleviate overfitting to the training distribution and leads to improvements on external datasets.
>
> For the experimental setup, we used 15,000 non-overlapping patches from the SIDD training data to generate synthetic noisy images and train the DnCNN denoising network. This approach ensures unique noisy samples without repetition and creates a controlled environment to systematically analyze the factors influencing performance on external datasets.
>
> As shown in the table below, we created three separate training groups: $\times$1, $\times$2, and $\times$4. These factors represent the multiplication applied to the original number of patches to generate synthetic noisy images using MFN for each clean image. Note that the performance on the in-distribution dataset (SIDD+) remains consistent. The denoising network demonstrates improved robustness across all real-world datasets as the number of samples increases, with $\times$4 producing the best average results. These findings suggest that utilizing multiple noise samples effectively reduces overfitting by ensuring sufficient noise diversity, thus highlighting the effectiveness of MFN.
>
> | DnCNN + MFN | PolyU | Nam | SIDD Validation | SIDD+ | Average |
> | --- | --- | --- | --- | --- | --- |
> | # Samples | PSNR↑ / SSIM↑ | PSNR↑ / SSIM↑ | PSNR↑ / SSIM↑ | PSNR↑ / SSIM↑ | PSNR↑ / SSIM↑ |
> | $\times$1 | 37.40/**0.9569** | 37.29/0.9578 | 36.55/0.8887 | 35.72/0.8997 | 36.74/0.9258 |
> | $\times$2 | 37.61/0.9546 | 37.92/**0.9580** | 36.95/0.8838 | 35.87/**0.9078**  | 37.09/0.9260  |
> | $\times$4 | **37.72**/0.9542 | **37.94**/0.9578  | **37.27**/**0.8955** | **36.00**/0.9022 | **37.23**/**0.9274**  |
>
> ---
>
> > **(Q1) Reason of Two-Stage Training Scheme.**
>
> Our framework consists of an autoencoder for latent code extraction and noisy image reconstruction, and a consistency model for latent code generation. During inference, the generated latent code is fed into the decoder to produce the final synthesized noisy image. Therefore, both reconstruction performance and generative capabilities are crucial for synthesizing high-quality noisy images. However, as stated in Sec. 4.2 of [1*], a joint training pipeline requires a delicate balance between reconstruction and generative capabilities. As a result, numerous experiments must be conducted to find the optimal weight, which is time- and resource-consuming. To address this issue, we propose a two-stage framework, where the reconstruction network is trained first and frozen while training the consistency model, ensuring training stability.
>
> ---
>
> >**Reference**
>
> [A] Fbi-denoiser: Fast blind image denoiser for poisson-gaussian noise, CVPR 2021
>
> [B] Practical blind denoising via swin-conv-unet and data synthesis, MIR 2023
>
> [C] NERDS: A General Framework to Train Camera Denoisers from Raw-RGB Noisy Image Pairs, ICLR 2023
>
> [1*] High-Resolution Image Synthesis with Latent Diffusion Models, CVPR 2022

---

> > ### Author Response · Authors · 2024-11-22
> >
> > Thank you for your valuable feedback and patience. We are pleased to inform you that the (W2) experiment is now complete, and we have updated the comment with the results. We look forward to any further feedback you may have.

---

> > > ### Comment · Reviewer_yT2B · 2024-11-27
> > >
> > > Thank you for your detailed responses. Your answers alleviated my concerns, and the revised version should include the experiments and discussions above.

---

> ### Author Response · Authors · 2024-11-27
>
> We would like to express our sincere gratitude to the reviewer for taking the time to evaluate our manuscript and for providing thoughtful and constructive feedback. The insightful comments and suggestions have been invaluable in enhancing our proposed methodology.
>
> In response to the reviewer’s feedback, we have revised the manuscript and have uploaded the revised version.
>
> We hope that these revisions have further strengthened the quality of our paper. Once again, we thank the reviewer for valuable feedback and for considering the changes we have made.

---

### Official Review · Reviewer_GejD · 2024-11-04

**Soundness:** 2
**Presentation:** 3
**Contribution:** 3
**Rating:** 5
**Confidence:** 5

**Summary:**

This paper proposes a metadata-free noise model to model real-world noise in denoising tasks without relying on camera metadata. The key idea is that the proposed method is composed of a Prompt Autoencoder and Prompt DiT, encodes input noise characteristics to generate realistic noisy images. Experimental results on various benchmark datasets verify MFN’s strong performance in noise generation and denoising.

**Strengths:**

- The motivation for MFN is clear, which successfully removes metadata dependency, offering a new approach for real-world noise generation.
- The framework is well-structured, and experiments across many denoising benchmarks validate its effectiveness in realistic noise generation and denoising.
- The paper is clearly writen and the overal explanation is logical.

**Weaknesses:**

- In Table 4, I am not sure how $\textit{Real}$ is trained. I mean if the data for training the proposed noise model covers the data for training $\textit{Real}$, the modeling-synthesizing-denoising pipeline becomes meaningless since the complex pipeline doesn't provide any performance gain over $\textit{Real}$ in terms of denoising results. This should be clarified with more detailed information.
- The noise distribution of sRGB images are very complex, it is signal-dependent, spatial-correlated, and also has consistent fixed patterns. How do the latent code and decoder handle this complexity to accurately model a signal-dependent, spatially correlated noise distribution?
- The entire pipeline is not clearly explained. Do we need to train a new PAE and P-DiT for each new camera? Additionally, I would like clarification on whether $n_{Real}$, $I_{Clean}$, and $I_{Noisy}$ are from the same scene. Specifically, is $n_{Real} = I_{Noisy} - I_{Clean}$ for both training and inference? While it makes sense to assume this during training, Figure 1(b) seems to imply the same notation in the reference, which raises questions about generating noisy images from the same noise. In addition, if $n_{Real}$ is derived from a different image captured with the same target camera and ISO, this approach may still be more challenging and less practical than directly obtaining ISO and camera metadata, as $n_{Real}$ would require paired noisy and clean images.
- I recommend that the authors compare their method with state-of-the-art self-supervised denoising methods such as LGBPN, as well as with supervised training results using the same data as the data for training the proposed noise model. (I am not sure whether this corresponds to $\textit{Real}$ in Tables 4 and 5.)

**Questions:**

In conditions where the ISO value and camera type is known in advance, will it performs better if we inject the ISO and camera information into the framework?

---

> ### Author Response · Authors · 2024-11-21
>
> Dear Reviewer GejD,
>
> We would like to thank the reviewer for recognizing our contribution and pointing out the specific strength of our method, as well as constructive concerns. We address the reviewer's concerns as follows.
>
> ---
>
> > **(W1) Clarification Regarding the Usage of Synthetic Data for Denoising**
>
> As presented in previous works [1*,2*,3*], the primary purpose of Table 4 in the main manuscript is to evaluate the quality of synthesized noise in comparison to real noise. Due to the complex properties of sRGB noise, surpassing the performance of models trained on real datasets using synthetic datasets alone remains a challenging task.
>
> However, combining real and synthetic datasets can enhance denoising performance, making the proposed MFN model more effective for practical applications. In the subsequent table, we adjust the ratio of the $Real$ dataset and the synthesized dataset. The results indicate that training the denoising network with a mixed dataset (50% Real / 50% Synthesized) provides performance gains over training with only the $Real$ data (100% Real / 0% Synthesized) across all datasets. Specifically, this configuration maintains strong performance across different camera sensor types (e.g., DSLR) on the PolyU and Nam datasets, while also outperforming the $Real$ (100% Real / 0% Synthesized) network on the SIDD Validation, SIDD+, and DND datasets.
>
> These results underscore leveraging both the existing training dataset and the synthesized dataset leads to improved denoising performance, showcasing the effectiveness of our approach.
>
> | MFN + DnCNN | SIDD Validation | SIDD+ | PolyU | Nam | DND | Average |
> | --- | --- | --- | --- | --- | --- | --- |
> | Real / Synthesized (%) | PSNR↑ / SSIM↑ | PSNR↑ / SSIM↑ | PSNR↑ / SSIM↑ | PSNR↑ / SSIM↑ | PSNR↑ / SSIM↑ | PSNR↑ / SSIM↑ |
> | 100 / 0 ($Real$ in Tab. 5) | 37.72 / 0.8905 | 35.68 / 0.8860 | 36.34 / 0.9204 | 35.35 / 0.8828 | 38.85 / 0.9434 | 36.79 / 0.9046 |
> | 0 / 100 (Ours in Tab. 5) | 37.64 / 0.8960 | 36.23 / 0.9072 | 37.93 / 0.9609 | 38.08 / **0.9630** | 38.75 / 0.9468 | 37.73 / 0.9348 |
> | 50 / 50 | **37.96** / **0.9047** | **36.57** / **0.9137** | **37.98** / **0.9610** | **38.09** / 0.9617 | **39.05** / **0.9472** | **37.93** / **0.9376** |
>
> ---
>
> > **(W2) Explanation of Modeling Signal-Dependent and Spatially Correlated Noise**
>
> The core concept of MFN is to embed noise-relevant representations into prompt features, thereby enriching the latent space with meaningful and discriminative information.
>
> To achieve this, we address the complexity of noise characteristics from both local and global perspectives as follows.
>
> (**Spatially Correlated Properties**) To capture the spatially complex and non-linear local patterns introduced by the ISP pipeline, we extract local correlation maps from real noise data. These maps encode spatial relationships and patterns, which are subsequently utilized to generate local prompt features.
>
> (**Signal-Dependent Properties**) To account for global noise characteristics, including variations in noise strength influenced by the ISP pipeline's signal gain, we compute statistical features, i.e. mean and standard deviation, from feature maps. These statistical features form the basis for global prompts, which encapsulate global noise characteristics. In the decoder, clean signals are conditioned to learn the signal dependencies of real-world noise through data-driven learning. This approach enables the decoder to model and reconstruct signal-dependent noise distributions.
>
> To evaluate the efficacy of the proposed method, we performed metadata classification, noise generation, and reconstruction experiments (please refer to Tables 6, 7, and 8 in the main manuscript). The results demonstrate that noise-relevant information is effectively captured and embedded within the prompt features, enabling the MFN to effectively handle the complexity of real-world noise distributions with high accuracy.

---

> ### Author Response · Authors · 2024-11-21
>
> > **(W3-i)** **Unified Modeling of Noise Across Multiple Cameras**
>
> Our proposed method does not require training a new Prompt Autoencoder (PAE) and Prompt-DiT (P-DiT) for each new camera. Instead, diverse noise characteristics from the training dataset are embedded into the unified prompt components. These components dynamically produce unique prompt features conditioned on the input noise, as described in Equations (5-8) in the main manuscript. This design enables the MFN framework to handle multiple noise types—defined by different camera models or characteristics—within a single unified model, offering significant efficiency and scalability for real-world applications.
>
> > **(W3-ii)** **Clarifying the Use of $n_{Real}$, $I_{Clean}$, and $I_{Noisy}$ Across Training and Inference**
>
> We use paired  $I_{Noisy}$  and  $I_{Clean}$  during both the training and testing phases to calculate $n_{Real}$ as  $n_{Real} = I_{Noisy} - I_{Clean}$. This paired setting ensures better performance. However, we acknowledge that it may be less practical in scenarios where paired data ($I_{Noisy}$  and  $I_{Clean}$) is unavailable.
>
> To address this limitation, we conducted additional experiments to evaluate MFN's performance in an unpaired setting. In this case, noise from the training dataset ($n_{Real}^{Meta} $), sampled using the target metadata setting (e.g., camera, ISO), is utilized to synthesize noisy images with unpaired  $I^{Test}_{Clean}$ at test time. The results on SIDD validation set, summarized in the table below, demonstrate that while the unpaired setting shows slight performance degradation compared to the paired setting, MFN still achieves comparable or better noise quality metrics (KLD and AKLD) relative to methods such as NeCA-W and NAFlow.
>
> This flexibility highlights MFN's adaptability. In the unpaired setting, MFN effectively synthesizes noisy images without requiring paired  $I_{Noisy}$  and  $I_{Clean}$, making it suitable for scenarios where paired datasets are not available. Additionally, in the paired setting, MFN demonstrates its capability to handle noise without relying on explicitly defined metadata such as ISO or camera type. These results reinforce MFN's practicality across a range of use cases, whether paired or unpaired data is available.
>
> | **Model** | **Given** $I_{Noisy}^{Test}$ **at test phase?** | **KLD↓** | **AKLD↓** |
> | --- | --- | --- | --- |
> | NeCA-W | x | 0.0342 | 0.1436 |
> | NAFlow  | o | 0.0305 | 0.1306 |
> | MFN (unpaired) | x | 0.0226 | 0.1223 |
> | MFN  | o | **0.0194** | **0.1163** |
>
> ---
>
> > **(W4) Application of Synthetic Data in Self-Supervised Denoising**
>
> As mentioned by the reviewer, we have also conducted experiments on the self-supervised denoising model, LGBPN [4*], in addition to the supervised denoising model. As discussed in Section W1, we trained LGBPN with the real SIDD data (100% Real / 0% Synthesized), fully synthesized data (0% Real / 100% Synthesized), and mixed data (50% Real / 50% Synthesized). The results show a similar trend to those in the table from Section W1, where adopting synthetic data for training helps improve performance across different real-world datasets. Specifically, using mixed data (50% Real / 50% Synthesized) dramatically increases performance across all real-world datasets compared to $Real$ (100% Real / 0% Synthesized), yielding the best averaged result.
>
> These results highlight that leveraging both the existing training dataset and the synthesized dataset leads to improved denoising performance, showcasing the effectiveness of our approach in both supervised and self-supervised settings.
>
> | MFN + LGBPN | SIDD Validation | SIDD+ | PolyU | Nam | DND | Average |
> | --- | --- | --- | --- | --- | --- | --- |
> | Real / Synthesized (%) | PSNR↑ / SSIM↑ | PSNR↑ / SSIM↑ | PSNR↑ / SSIM↑ | PSNR↑ / SSIM↑ | PSNR↑ / SSIM↑ | PSNR↑ / SSIM↑ |
> | 100 / 0  | 35.99/0.8630 | 35.23/0.8975 | 37.01/0.9517 | 37.19/0.9566 | 37.36/0.9266 | 36.56/0.9191  |
> | 0 / 100 | 35.71/0.8630 | **35.76**/**0.9164** | 37.31/0.9542 | 37.42/0.9570 | 37.35/0.9250 | 36.71/0.9231  |
> | 50 / 50 | **36.94**/**0.9019** | 35.56/0.9152 | **37.44**/**0.9570** | **37.76**/**0.9621** | **38.29**/**0.9412** | **37.20**/**0.9355**  |

---

> > ### Comment · Reviewer_GejD · 2024-11-25
> >
> > During the first review round, I found the concept of “metadata-free” noise modeling very impressive. Existing methods typically require additional information, such as camera type and ISO, for noise model learning and generation. I initially thought this paper addressed this issue to enhance practicality, as suggested by the term “metadata-free.” However, some critical details were missing in the initial submission, making it difficult to fully assess its feasibility.
> >
> > After reading the authors’ clarification in the response, I now find that the practicality of this approach is even lower than other noise modeling methods that require metadata. This is because the proposed method $\textbf{relies on paired data not only for training but also for evaluation}$. I cannot accept the use of paired data during evaluation, as it raises a fundamental question: if real paired data is already available, why should we use it to embed synthetic noise and then evaluate performance on this data? This approach seems redundant and undermines the practicality of the method.
> >
> > Another issue I raised during the review remains unresolved. The noise model is trained on paired data to synthesize noisy images, which are then used for training a denoiser. However, direct training of the denoiser using the same paired data (without going through the noise synthesis process) achieves better performance. In other words, the proposed noise synthesis method actually reduces denoising performance.
> >
> > While the authors have provided additional efforts to address these two issues, these additional approaches and experiments represent new contributions rather than clarifications or verifications of the claims made in the submission. As these solutions were not part of the original paper, they cannot be considered in the evaluation of the current submission.
> >
> > Given these concerns, I have decided to lower my rating.

---

> ### Author Response · Authors · 2024-11-21
>
> > **(Q1) Effect of Explicit Metadata Conditiong on P-DiT**
>
> As recommended by the reviewer, we examined the effectiveness of incorporating explicit metadata for MFN conditioning. In the table below, we present an ablation study on four different conditioning types (non-conditioning, metadata-only, prompt-only, and combined). Each model was trained with a distinct conditioning configuration.
>
> Our findings indicate that using metadata alone for conditioning yields inferior performance compared to using only the prompt. However, when metadata is available, incorporating additional metadata alongside the prompt features achieves competitive performance in terms of both KLD and AKLD.
>
> Given that metadata is often unavailable in practical scenarios (L48 in the main manuscript), we propose the MFN framework, which demonstrates competitive generation performance without relying on metadata.
>
> | MFN Conditioning |  | SIDD validation |  |
> | --- | --- | --- | --- |
> | Prompt | Metadata | KLD**↓** | AKLD**↓** |
> | x | x | 0.5661 | 0.4132 |
> | x | o | 0.0267 | 0.1285 |
> | o | x | 0.0261 | **0.1108** |
> | o | o | **0.0215** | 0.1169  |
>
> ---
>
> > **Reference**
>
> [1*] Modeling sRGB Camera Noise With Normalizing Flows, CVPR 2022
>
> [2*] sRGB Real Noise Synthesizing with Neighboring Correlation-Aware Noise Model, CVPR 2023
>
> [3*] sRGB Real Noise Modeling via Noise-Aware Sampling with Normalizing Flows, ICLR 2024
>
> [4*] LG-BPN: Local and Global Blind-Patch Network for Self-Supervised Real-World Denoising, CVPR 2023

---

> ### Author Response · Authors · 2024-11-26
>
> > **(A1) Practicality of the Meta-Free Method**
>
> We appreciate the reviewer’s thoughtful feedback and would like to clarify the advantages of metadata-free noise modeling and address the concerns regarding the use of paired data during evaluation.
>
> Metadata-based noise modeling methods, such as NeCA-W, synthesize noise using metadata and clean images. While this approach eliminates the need for paired noisy-clean data, it introduces challenges in generalization. For instance, NeCA-W requires separate, camera-specific models, with five distinct models needed to represent the SIDD dataset, which limits its practicality on unseen datasets. Furthermore, it exhibits inferior performance compared to methods like NAFlow and ours, particularly in terms of generalization and robustness.
>
> Although our method requires paired data for evaluation in terms of KLD and AKLD, the ultimate goal of noise generation is to enable robust denoising performance through training data augmentation. While our method requires paired inputs for noise generation, it is important to note that the denoising results presented in Table 4 and Table 5 were achieved using datasets synthesized exclusively from the paired SIDD training dataset for training the denoising networks.
>
> In other words, all existing methods for denoising with synthetic training images ultimately rely on the same paired datasets. Notably, unlike our approach, other noise modeling methods additionally leveraged metadata. We emphasize this critical distinction to highlight the uniqueness of our method.

---

> ### Author Response · Authors · 2024-11-26
>
> > **(A2) Motivation of Noise Synthesis Process**
>
> The primary challenge addressed in this work is the difficulty of collecting large-scale, diverse real-world datasets. Gathering such datasets is labor-intensive, time-consuming, and costly. Moreover, the limited size and diversity of these datasets often lead to overfitting due to constrained variance in their distributions, as evidenced by the $Real$ results in Table 4.
>
> To address this issue, we propose MFN that learns the noise distribution from the training dataset. This framework enables the generation of an infinite number of noise samples from the learned distribution, effectively expanding the diversity of the training data without using metadata. This approach mitigates overfitting and improves the robustness of the denoising network, as demonstrated in Table 5.
>
> However, in Table 4 of the main manuscript, the reviewer observes that training the denoising network exclusively on the synthesized dataset leads to slightly degraded performance on the SIDD dataset. Despite the slight degradation in performance when using the synthesized dataset generated by MFN, our method significantly outperforms recent state-of-the-art (SOTA) models, such as NeCA-W and NAFlow, achieving only a **0.08 dB** gap in PSNR while surpassing SSIM by **0.001** compared to the $Real$ dataset.
>
> We note that, in Table 5 of the main manuscript, we observe that the denoising network trained exclusively on the $Real$ dataset exhibits overfitting issues with the SIDD dataset. This overfitting leads to reduced robustness across various other real-world datasets compared to the MFN approach, thereby limiting the ability to achieve performance that surpasses the $Real$ results.
>
> To further enhance the proposed MFN model's effectiveness for practical applications "without any modification of MFN", combining real and synthetic datasets generated by MFN can significantly improve denoising performance, yielding the best-averaged score across various real-world datasets.
>
> We argue that this mixed dataset experiment should be considered a key part of the evaluation for the current review process. While it does not introduce additional contributions to the framework itself, the only change made in this experiment is the training dataset setting (combining existing datasets).
>
> Furthermore, the ICLR review process allows authors to revise their paper with new information, and the official AC guidelines on the ICLR website explicitly state: "The final decision and meta-review should take the revisions into account.”

---

> ### Author Response · Authors · 2024-11-26
>
> > **(A3) Consideration of Additional Experiments for Evaluation in the Current Submission**
>
> As the ICLR review process allows authors to revise their paper with new information, and the official AC guidelines on the ICLR website explicitly state, "The final decision and meta-review should take the revisions into account.", we have added the results of additional experiments to the appendix of the revised main script to address the concerns raised by the reviewer during the discussion. Moreover, since we only modified how the datasets are constructed—such as adjusting the ratio between original and synthetic noisy images—for training the denoising network and did not alter the core methodology of MFN while obtaining the additional experimental outcomes, these experiments do not constitute new contributions. Instead, they serve to further clarify the proposed task of data augmentation, which can be applied in scenarios where a limited number of real-world paired noisy-clean images is available.

---

> ### Author Response · Authors · 2024-12-02
>
> Dear Reviewer GejD,
>
> Thank you for your valuable contribution to the development of our paper.
>
> As the discussion period draws to a close, we kindly look forward to your response.
>
> Please let us know if you have any remaining concerns or if there is anything further we can clarify to assist you.

---

### Meta-Review · Area_Chair_b7Nv · 2024-12-24

**Metareview:**

This paper aims to synthesize realistic noisy images without relying on metadata, addressing the challenge of obtaining paired noisy-clean images. The proposed techniques, including PAE and P-DiT, are conceptually sound, and experimental results demonstrate that a denoising model trained on the synthesized data outperforms models based on other noise synthesis methods. However, from a practical perspective, Reviewer kgBq highlighted a critical limitation: the proposed method requires real noise during inference. Given such paired real noisy-clean images, one could directly train a denoising network, potentially negating the necessity of the proposed approach.  Reviewer GejD echoed this concern during the author-review discussion.

In response, the authors conducted new experiments and proposed two new approaches to address this issue:
(i) Enhancing real image pairs by incorporating synthetic data, with a 50-50 mix ratio appearing optimal.
(ii) Extending MFN to MFN(unpaired), albeit with some performance loss.
The authors argued that these new proposals should be considered when evaluating the revised paper.

Under ICLR25 regulations, AC agrees that new experiments and proposals should be factored into the evaluation. However, upon further consideration, the new proposals introduce potential concerns and confusion:

For proposal (i), the results of mixing Real+MFN are presented, but it raises the question of how mixing real images with synthetic data from other methods would perform. It’s conceivable that an alternative mix might surpass Real+MFN. Furthermore, if mixed real+synthesis yields the highest performance gain, the paper's motivation and framing should be revised to reflect this finding. Additionally, since real paired data are required for this approach, the original motivation—to avoid metadata dependency—could be undermined, as obtaining metadata is generally not more challenging than capturing paired data. For proposal (ii), the use of metadata in MFN(unpaired) directly contradicts the paper's stated "metadata-free" objective.

In summary, Reviewers kgBq and GejD identified a key limitation that significantly impacts the method's practical applicability. Despite the authors’ efforts to address this issue through new proposals and experiments, these solutions introduce further complications and inconsistencies. AC believes that without a substantial rewrite of the paper—particularly the motivation—and a comprehensive reorganization of the experiments, the manuscript, even in its revised form, is likely to confuse readers. Consequently, AC concurs with the recommendations of Reviewers kgBq and GejD and leans toward rejecting this paper.

**Additional Comments On Reviewer Discussion:**

Two reviewers rated the paper as 8-Accept, albeit with mild confidence scores. In contrast, two experienced reviewers, kgBq and GejD, provided ratings of "Marginal Reject" and "Reject," both with the highest level of confidence. AC noted that Reviewer GejD lowered the score after recognizing that paired real data are required for both training and inference stages of the proposed method.

Although there is some disagreement regarding whether the new experimental results—covering a substantial scope—should be considered, AC reviewed the guidelines and agrees that these results and proposals should be factored into the evaluation. However, even with the new results and proposals, AC believes the paper introduces new concerns and potential confusion.

The authors are encouraged to reflect on all feedback provided during this review process and submit a more refined version to a future venue. In AC's opinion, just emphasizing MFN(unpaired) with metadata could lead to a coherent and high-quality paper.

---

### Decision · Program_Chairs · 2025-01-22

Reject